# AutoElicit: Using Large Language Models for Expert Prior Elicitation in Predictive Modelling

Alexander Capstick [1 2]   Rahul G. Krishnan [3 4]   Payam Barnaghi [1 2 5]

## Abstract

Large language models (LLMs) acquire a breadth of information across various domains. However, their computational complexity, cost, and lack of transparency often hinder their direct application for predictive tasks where privacy and interpretability are paramount. In fields such as healthcare, biology, and finance, specialised and interpretable linear models still hold considerable value. In such domains, labelled data may be scarce or expensive to obtain. Well-specified prior distributions over model parameters can reduce the sample complexity of learning through Bayesian inference; however, eliciting expert priors can be time-consuming. We therefore introduce AutoElicit to extract knowledge from LLMs and construct priors for predictive models. We show these priors are informative and can be refined using natural language. We perform a careful study contrasting AutoElicit with in-context learning and demonstrate how to perform model selection between the two methods. We find that AutoElicit yields priors that can substantially reduce error over uninformative priors, using fewer labels, and consistently outperform in-context learning. We show that AutoElicit saves over 6 months of labelling effort when building a new predictive model for urinary tract infections from sensor recordings of people living with dementia.

[1]Department of Brain Sciences, Imperial College London [2]UK Dementia Research Institute, Care Research and Technology Centre [3]Vector Institute [4]Department of Computer Science, University of Toronto [5]Data Research, Innovation and Virtual Environments (DRIVE) Unit, The Great Ormond Street Hospital. Correspondence to: Alexander Capstick <alexander.capstick19@imperial.ac.uk>.

*Proceedings of the $42^{nd}$ International Conference on Machine Learning*, Vancouver, Canada. PMLR 267, 2025. Copyright 2025 by the author(s).

## 1. Introduction

Labelling data in medicine can be a costly and time-consuming undertaking (Ghassemi et al., 2020; Cooper et al., 2023). We are particularly interested in the development of automated tools to assist with healthcare monitoring for people living with dementia. In our study, participant data is recorded through passive monitoring devices. Although input features are continuously collected at scale, the process of labelling adverse health conditions, such as urinary tract infection (UTI), can be challenging. Expert knowledge of a task, as privileged information (Vapnik et al., 2015), can provide prior distributions of model parameters, which often reduces the number of samples needed to learn performant models while also enabling more accurate estimates of predictive uncertainty. However, this expert knowledge about model parameters can be difficult to obtain due to the challenges of performing expert prior elicitation, particularly in low-resource settings (Mikkola et al., 2024).

Linear models are widely used in real-world applications, especially in healthcare and as part of our wider study on dementia care, due to their interpretability, robustness, and clinical familiarity (Capstick et al., 2024; Tennant et al., 2024; Ramírez Medina et al., 2025). Many studies do not use prior distributions on their models, possibly due to the challenges associated with elicitation, but would likely benefit greatly from them.

Due to their diverse training data and rate of development (Naveed et al., 2023), large language models (LLMs) offer a new opportunity to fully or partially automate the prior elicitation process, especially given recent advances in reasoning methods (Huang & Chang, 2023). By prompting an LLM to approximate a specific form of expertise in a given field, it is possible to extract insights about a predictive model's prior based on the LLM's training data. This motivates the construction of a probabilistic framework to extract prior task knowledge from language models in the form of distributions over the parameters of a linear predictive model. AutoElicit combines developments in LLMs with trusted and robust linear predictive models, that are already widely understood and utilised in real-world applications.

In-context learning has been studied as a way to use LLMs

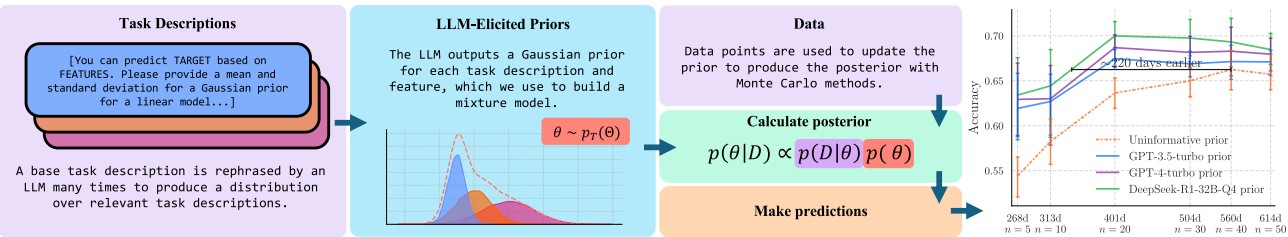

*Figure 1.* **AutoElicit: Prior elicitation using a language model.** The left-hand figure demonstrates the process, whilst the right-hand figure illustrates the benefits of using AutoElicit, achieving the same peak accuracy after $n \approx 15$ labels, 220 days earlier in the study.

for predictive tasks (Dong et al., 2024). Here, LLMs are given a task description and demonstration examples and asked to make predictions on unseen data directly (Brown et al., 2020). There are several reasons why this prompted prediction may not suffice. First, LLMs can incur a large computational cost to run, rendering their use in high-frequency or low-resource settings unlikely. Second, applications may require arithmetic operations within models, an area where LLMs continue to struggle (Spithourakis & Riedel, 2018). Third, LLMs, unless explicitly trained on a specific predictive problem, lack transparency and may not perform Bayesian inference (Min et al., 2022), which can lead to unreliable posterior updates and predictions.

Our work makes the following contributions:

- We introduce AutoElicit, an algorithm to generate LLM-elicited priors for linear models. We find using LLMs for expert prior elicitation (Figure 1) provides higher posterior accuracy and lower mean squared error than uninformative priors on various numerical predictive tasks, improving models with fewer examples.

- To compare prior elicitation with in-context learning (ICL), we develop an approach to extract the implicit priors used in ICL, which we find to be less informative. To further understand the differences, we study whether an LLM performs Bayesian inference during ICL. We extract the in-context posterior distribution and measure its divergence from posterior Monte Carlo samples of the extracted prior. We find that the language model inconsistently approximates Bayesian inference on real-world datasets.

- We study when ICL is better than AutoElicit and show how to use the Bayes factor to guide model selection, finding that LLM-elicited priors are the preferred choice for all tasks tested. When predicting UTIs in our study on dementia care, we find that AutoElicit priors result in performant accuracy over six months earlier than with uninformative priors (Figure 1).

Our results provide insight on when and how to use LLMs to build better linear models, with fewer samples. Open-source code[1] to reproduce our work is described in Appendix A.1.

## 2. Related Work

Garthwaite et al. (2005) and O'Hagan et al. (2006) discuss eliciting prior distributions from human experts using a standardised procedure: Experts are trained in probabilities to provide judgements of a task, which can be used to fit probability distributions. Later, Oakley & O'Hagan (2019) provide software and an e-learning course, which trains human experts in the knowledge required to describe informative priors. However, this can be challenging, costly, or infeasible in low-resource settings (Mikkola et al., 2024).

In parallel, research shows that LLMs encode varied domain knowledge, which could be extracted for predictive tasks, possibly in discussions with experts (Gao et al., 2024). Recent works have used LLMs for prior elicitation in multiple ways. In data pre-processing, Choi et al. (2022) and Hollmann et al. (2024) show that LLMs can perform feature selection and engineering respectively, whilst Zhang et al. (2023) and Zhang et al. (2024) use LLMs to specify a modelling pipeline given a task description. This suggests language models might also be able to provide information over the parameters of predictive models directly. In this direction, Gouk & Gao (2024) use LLMs to generate synthetic data from public datasets, which define priors on linear models through a separate likelihood term. However, they do not consider that these datasets might have appeared in the LLM's training (Appendix A.8 and A.9.5), making it challenging to evaluate. Related, Selby et al. (2025) evaluate LLMs for estimating correlations between features and targets, and Zhu & Griffiths (2024) use them to elicit priors for causal learning, proportion estimation, and everyday quantities. However, this knowledge is not used for predictive models. Further related work is discussed in Appendix A.2.1. In contrast, we provide a framework for using LLMs to elicit priors over the parameters of linear models directly, which we evaluate on public and private predictive tasks.

---

[1] https://github.com/alexcapstick/llm-elicited-priors

LLMs can also be used to produce posterior predictive distributions. In-context learning is an active research area, mainly focused on natural language tasks (Dong et al., 2024), with many works aiming to explain the underlying mechanisms. Relevant to us, Requeima et al. (2024) prompt LLMs with task descriptions and data to generate posterior predictive distributions for regression tasks. Additionally, Hollmann et al. (2022; 2025) train a transformer to solve synthetically generated tasks from a prior on tabular datasets and use in-context learning at prediction time. However, these methods sample LLMs for every prediction, which is resource-intensive and potentially unreliable. Additional related work is given in Appendix A.2.2. In our work, we propose a strategy to approximate the in-context prior and posterior distributions when imitating a linear model, which we compare to LLM-elicited priors. Using these, we empirically evaluate whether in-context learning approximates Bayesian inference before using model selection.

LLMs have been evaluated for use in healthcare (Thirunavukarasu et al., 2023), demonstrating a broad knowledge of clinical problems and strong performance on medical questions, even without fine-tuning (Nori et al., 2023; Singhal et al., 2023). Despite this, their direct use for decision making, through in-context learning, is not recommended as they sometimes produce inaccurate information. However, linear models have found wide and trusted uses in clinical applications (Ramírez Medina et al., 2025; Tennant et al., 2024), since their model weights have intuitive descriptions. This motivates us to combine LLMs, which can reason about clinical problems and risk factors, with linear models, which are interpretable and familiar to clinicians.

## 3. Background

### 3.1. Prior Elicitation with Experts

Bayesian inference incorporates knowledge about a predictive task before data is observed (Murphy, 2022). A prior $p(\theta)$ is placed on a predictive model, which holds intuitions about the parameters $\theta$. Well-specified priors allow us to estimate predictive uncertainty through the posterior predictive distribution and reduce the number of data points needed to construct comparably accurate predictive models.

Prior elicitation constructs prior distributions on model parameters, typically through expert judgement (Oakley & O'Hagan, 2019; Gosling, 2018). This can be resource-intensive and requires access to one or more domain experts who have been trained in prior elicitation (Mikkola et al., 2024).

### 3.2. Linear Models

Because of their simplicity, linear models have interpretable parameter weights, making them trusted in clinical contexts.

Model weights are explicitly understood in their effect on predictions, as they multiply directly with features, making them ideal to use with prior elicitation in healthcare.

### 3.3. In-context Learning

In-context learning uses the knowledge that LLMs acquire during training to make predictions on unseen data. Here, LLMs attempt to learn a function for predicting unseen data based on a task description and demonstration examples.

Let the space of all predictive tasks be $\mathcal{T}$, a single classification or regression task be $T \in \mathcal{T}$, and the space of possible LLM prompts be $\mathcal{I}$. Given an LLM $M$ and $K$ task descriptions $I_k \sim p(I|T) = p_T(I)$, a prediction of $\tilde{y}$ on features $\tilde{x}$ using in-context learning with training data $\mathcal{D}$ is:

$$p_{M,T}(y|\tilde{x}, \mathcal{D}) = \sum_{k=1}^{K} p_{M,T}(\tilde{y}|\tilde{x}, \mathcal{D}, I_k) p_{M,T}(I_k)$$

Although the mechanisms of in-context learning are under active research, we denote its implicit parameters as $\varphi$, implied by a task description and dataset:

$$p_{M,T}(y|\tilde{x}, \mathcal{D})$$
$$= \int_{\Phi} \sum_{k=1}^{K} p_{M,T}(y|\tilde{x}, \varphi) p_{M,T}(\varphi|\mathcal{D}, I_k) p_{M,T}(I_k) d\varphi \quad (1)$$

Here $\varphi$ is itself described by a prior distribution $p_{M,T}(\varphi|I_k)$ and a posterior distribution $p_{M,T}(\varphi|\mathcal{D}, I_k)$. $\varphi$ is latent, since in-context learning only returns $\tilde{y} \sim p_{M,T}(y|\tilde{x}, \mathcal{D}, I)$.

## 4. Methods

### 4.1. AutoElicit: Using Language Models to Elicit Priors

We aim to make expert prior elicitation more accessible by using the knowledge of LLMs in robust and clinically trusted predictive models. This motivates us to develop a framework for eliciting general prior distributions for linear models from LLMs, AutoElicit (Figure 1), which often have both technical knowledge of probability distributions and domain knowledge of a task. Our goal is to improve model performance with less data compared to an uninformative prior, whilst allowing model parameters to be interpretable.

Given an LLM $M$ and task $T$, we obtain a single Gaussian prior for each feature, for each task description $I_k$ by asking the LLM to provide its best guess of the mean and standard deviation: $(\mu_k, \sigma_k) \sim p_{M,T}(\mu, \sigma|I_k)$. Here, we use $K = 100$ task descriptions, produced by asking an LLM to rephrase one human-written "system" and "user" role 10 times and taking their product (Appendix A.7.3).

Taking a mixture over the task descriptions, we construct a

prior $p_{M,T}(\theta)$ over linear model parameters $\theta$:

$$p_{M,T}(\theta) = \sum_{k=1}^{K} \pi_k \mathcal{N}(\theta | \mu_k, \sigma_k{}^2)$$

$$\text{where } (\mu_k, \sigma_k) \sim p_{M,T}(\mu, \sigma | I_k) \text{ and } \pi_k \sim \text{Dir}(1) \quad (2)$$

Mixtures of Gaussians approximate any distribution with enough components, allowing us to fully capture the LLM's prior belief. We also expect that LLMs have learnt how to parametrise Gaussian distributions, although our framework does allow for other mixtures. In our work, this mixture has 100 components, using as many task descriptions, and in Appendix A.12 we find diminishing differences in the LLM-elicited priors as we increase the number of descriptions.

This $p_{M,T}(\theta)$ is the prior that we use for our linear models and is an alternative to one elicited from experts. However, additionally including expert information $E$ in the prompts can provide yet more informative priors: $p_{M,T}(\theta | E)$. This way, AutoElicit also supplements a typical prior elicitation process but only requires experts to provide knowledge as natural language, likely needing less training and cost. This approach enables the use of an LLM's knowledge in new environments, with more controls and transparency.

The natural comparison for AutoElicit priors is an uninformative prior. However, since we are using an LLM's prior knowledge and training data to make new predictions, this is also an alternative to in-context learning. The difference being that prior elicitation uses a marginal distribution and Bayesian inference, which address some of the shortfalls of in-context learning on numerical predictive tasks.

## 4.2. In-context Learning

To compare in-context learning and AutoElicit, we must consider: (1) Does the LLM use the same elicited prior in-context? and (2) Does in-context learning approximate Bayesian inference? To answer these, we extract the prior and posterior distributions of the in-context model.

EXTRACTING THE IN-CONTEXT PRIOR AND POSTERIOR

We can write the implicit in-context model as: $f_\varphi : X \rightarrow Y$ where $f$ represents the class of the predictive model, which we specify in the prompt, $\varphi$ is the random variable associated with its parameters, and $X$ and $Y$ correspond to its domain and range. Based on the model class $f$ corresponds to, $y \in Y$ will either be a predicted target or its probability.

In-context learning returns a $y$ given an $x$. Since we do not have access to $\varphi$ directly, we propose to use $p_{M,T}(y | x, \varphi)$ and maximum likelihood estimation (MLE) to generate a sampling distribution for $p_{M,T}(\varphi)$ and $p_{M,T}(\varphi | \mathcal{D})$.

To do this, given a language model $M$ and task $T$, we construct the sample distribution (Murphy, 2022):

$$\varphi_{\text{MLE}} \sim \text{MLE}_{M,T}(f_\varphi(X) | X, I) \quad (3)$$

Where $\text{MLE}_{M,T}(f_\varphi(X) | X, I)$ refers to the MLE of the parameters $\varphi$ given input $X$, in-context predictions $f_\varphi(X)$, and task description $I$. This provides a sample of the estimated prior on $\varphi$ for the domain of $X$. Further, additionally providing training data $\mathcal{D}$, returns posterior samples:

$$\varphi_{\text{MLE}} | \mathcal{D} \sim \text{MLE}_{M,T}(f_\varphi(X) | X, \mathcal{D}, I) \quad (4)$$

In our experiments, for each $\varphi_{\text{MLE}}$, we sample 25 data points from $X \sim \mathcal{U}[-5, 5]$, five times for each of the 100 task descriptions. As the real data is normalised, these samples of $X$ will cover the domain of $f_\varphi$ with a high probability. Appendix A.3 provides more details on the process.

When specifying linear $f_\varphi$, we calculate the MLE in closed form as we have $y = \varphi_0 + \varphi \cdot x$, which is either a regression label or the logit of a class. Therefore, large approximation errors will be due to the LLM not using a linear $f_\varphi$.

By repeated sampling, we form distributions that estimate $p_{M,T}(\varphi)$ and $p_{M,T}(\varphi | \mathcal{D})$. These are over the hidden parameters of the in-context model and give us new visibility into the LLM's predictive process, assuming the model class can be specified in a prompt (Appendix A.4). Using these, we can test whether an LLM is using the elicited prior in-context and if it is approximating Bayesian inference.

IS THE LLM USING THE ELICITED PRIOR IN-CONTEXT?

To quantify the difference between the in-context prior ($p_{M,T}(\varphi)$) and the one we elicit ($p_{M,T}(\theta)$), we will use the energy statistic (Székely & Rizzo, 2013). This takes values $\in [0, 1]$ with 0 indicating that the two distributions are equal (Appendix A.5) and evaluates how consistently the LLM uses its background knowledge.

THE IN-CONTEXT POSTERIOR

We also test whether the LLM is approximating Bayesian inference during in-context learning. We use the approximated in-context prior $p_{M,T}(\varphi)$ and Monte Carlo methods (Hastings, 1970; Hoffman et al., 2014) (Appendix A.15) to sample a posterior, and compare it to the extracted in-context posterior $p_{M,T}(\varphi | D)$. The energy statistic quantifies the difference in these distributions. If they differ, it suggests the LLM is not performing Bayesian inference in-context and is not acting predictably after observing data from $\mathcal{D}$.

## 4.3. The Bayes Factor for Model Selection

If the LLM is approximating Bayesian inference in-context, we use the Bayes factor for model selection between in-context learning and AutoElicit priors. This calculates the

marginal likelihood ratio of two statistical models to determine under which the data was likely generated (Appendix A.6), defining a model selection strategy given any dataset.

For AutoElicit (method $\alpha_0$), we approximate the marginal likelihood by sampling the prior predictive log-likelihood over $\mathcal{D}$. We sample 500 predictions for 25 data points to get samples of $p_{M,T}(\mathcal{D}|\alpha_0)$. For in-context learning (method $\alpha_1$), we approximate $p_{M,T}(\mathcal{D}|\alpha_1)$ as the LLM provides predictions on features directly. We prompt the LLM with 100 task descriptions and ask it to predict on 25 data points from the dataset. This is repeated over 5 splits of the datasets. Further details are given in Appendix A.7.5.

## 5. Results

### 5.1. Experimental Setting

We present results on one synthetic and five clinical datasets, covering classification and regression (Appendix A.7.1).

The synthetic data set contains 3 features from $X \sim \mathcal{N}(0, 1)$ and targets calculated using $y = 2x_1 - x_2 + x_3 + \epsilon$, with $\epsilon \sim \mathcal{N}(0, 0.05)$. When describing this task to the LLM, we fully specify the relationship between the features and targets to control for the completeness of the descriptions.

We test four publicly available datasets. The Heart Disease dataset (Detrano et al., 1989): predicting a heart disease diagnosis from physiological features; the Diabetes dataset (Efron et al., 2004): predicting disease progression one year after baseline; the Hypothyroid dataset (Quinlan, 1986; Turney, 1994): predicting hypothyroidism based on biomarkers; and the Breast Cancer dataset (Wolberg et al., 1995): predicting the diagnosis of a breast mass.

We also evaluate AutoElicit on a *private* dataset not in the LLM's training, collected as part of our study on Dementia care. This contains 10 features of daily in-home activity and physiology data, and clinically validated labels of positive or negative urinary tract infection (UTI), based on the analysis of a urine sample by a healthcare professional. This dataset contains 78 people and 371 positive or negative days of UTI.

To explore if the LLM has memorised this data, we use two tests (Bordt et al., 2024) in Appendix A.8. Appendix A.7.6 explains how to use these methods on a new dataset.

All experiments in this section were conducted with OpenAI's GPT-3.5-turbo and GPT-4-turbo, and DeepSeek-R1-32B with 4-bit quantisation. All details are given in Appendix A.7, and other LLMs are tested in Appendix A.16.

### 5.2. How Informative are AutoElicit Priors?

We compare the posterior performance of a predictive model using an uninformative prior $\theta \sim \mathcal{N}(\theta|0, 1)$, with those elicited from three LLMs. Figure 2 shows that Au-

toElicit priors provide substantially better accuracy and mean squared error (MSE) than an uninformative prior, and at least one of the LLMs provides the greatest performance for all training sizes and tasks. The complete accuracy distributions and a detailed analysis is given in Appendix A.9; Appendix A.10 describes the associated cost; and A.12 studies how the number of descriptions affects the priors.

When eliciting priors for $y = 2x_1 - x_2 + x_3$, we specify the equation in the prompt to test the LLMs' ability to provide priors given complete descriptions (incomplete descriptions are studied in Appendix A.11). In Figure 2, the LLM-elicited priors provide an order of magnitude lower test MSE compared to an uninformative prior after 5 observed examples and an equivalent MSE after 20, as the task was simple enough to be learnt with only a few examples. GPT-3.5-turbo provided the best prior as it returned sharp distributions (Appendix A.9.2), reflecting the true parameter values. This shows LLMs can combine their knowledge of probability with task descriptions to form useful priors.

For Heart Disease, uninformative and GPT-3.5-turbo-elicited priors produced the same mean posterior accuracy, as this elicited prior contains many standard Normal distributions (Figure 18). This is reassuring, as GPT-3.5-turbo responded with an uninformative prior when unsure of its knowledge. However, GPT-4-turbo and DeepSeek-R1-32B-Q4, returned considerably better priors. The appropriate prior can be chosen with the Bayes factor (Appendix A.17).

Of particular note are the results on the UTI task, a private dataset where all AutoElicit priors provide greater accuracy over the uninformative prior for all training sizes. Figure 1 shows these results considered alongside label collection time, illustrating that we achieve greater accuracy earlier in the study, accelerating model development and validation. This is discussed further in Appendix A.9.3.

Also of importance are the results on the classification tasks of the priors elicited from DeepSeek-R1-32B-Q4, a 4-bit quantised version of the distilled DeepSeek-R1 model. This suggests the reasoning qualities of more recently released LLMs are of significant use for eliciting priors.

In Appendix A.9.5, we also compare our results to those achieved using the approach presented by Gouk & Gao (2024). This method uses an LLM to generate feature values before labelling them with zero-shot in-context learning. These generated samples are provided alongside real data during the training of a linear predictive model. Therefore, the method presented in Gouk & Gao (2024) is susceptible to LLM data memorisation, putting it at risk of artificially providing improved posterior results on public datasets. In light of this, Figure 13 shows the results split by those on private and public tasks. Our approach yields improved posterior accuracy and mean squared error over this baseline.

*Figure 2.* **Test performance of the posterior distribution for varied training data sizes.** The average and 95% confidence interval of the mean posterior accuracy or Mean Squared Error (MSE) of 10 splits of the dataset using AutoElicit priors and an uninformative prior ($\theta \sim \mathcal{N}(\theta|0, 1)$). These are calculated on a test set of each dataset, with the green arrow pointing in the direction of metric improvement.

In Appendix A.13, we demonstrate how the priors elicited can be useful for explaining the final predictive model.

These results illustrate the remarkable effectiveness of using AutoElicit priors, which considerably improve posterior predictions, particularly for smaller training sizes, where label collection might have been challenging. Our framework makes expert prior elicitation more accessible.

### 5.3. Can Experts Supplement the Process?

Including expert information in the task description allows AutoElicit priors to combine learnt insights with domain knowledge. In the UTI task description, we now list three features that positively correlate with the label (Capstick et al., 2024): "It is known that the more previous UTIs a patient had, the more likely they are to have a UTI in the future. Someone with a UTI will also wake up more during the night. Someone with a UTI will have a greater bathroom usage."

Figure 3 shows that this information is incorporated when LLMs provide priors, as these three features become more positively predictive of a UTI. GPT-3.5-turbo had already identified this relationship, shown by the positive median of the priors, which explains the small improvement in posterior accuracy this additional information provides. Further, Appendix A.11 studies the priors elicited as we incrementally improve the description of a synthetic task, illustrating that GPT-3.5-turbo produces reliable distributions from text.

AutoElicit can therefore also be used to supplement typical prior elicitation methods, with the benefit that experts can convey their knowledge as natural language.

### 5.4. The In-context Priors and Posteriors

Figures 4 and 5 show GPT-3.5-turbo's in-context prior and posterior distributions approximated with maximum likelihood estimation (MLE). This allows us to study the implicit parameters used by the LLM to make predictions in-context.

The estimated priors in Figure 4 show variations in their values across the datasets, suggesting the LLM uses its own

knowledge in-context before observing demonstrations. The low MSE on the real-world tasks indicates that the LLM was reliably imitating a linear model, allowing us to achieve a low approximation error. However, the priors are not particularly informative, as they are often centred close to 0.

The estimated in-context posteriors in Figure 5 differ from the priors, suggesting the LLM has updated its knowledge based on demonstrations. For the Diabetes and Heart Disease tasks, the LLM mostly continued to produce linear predictions, and so our error increases, but remains low. However, the Breast Cancer and Hypothyroid dataset produces a large approximation error, as the LLM struggles to imitate a linear model. The approximation error on the synthetic task is discussed in Appendix A.4. We do not estimate the in-context posterior for the UTI dataset as it is private, and so we are unable to provide demonstrations.

In general, we better approximate the LLM's in-context prior than posterior. This is likely due to limitations in the LLM's capacity to interpret demonstrations for training, alongside instructions to provide linear predictions, in the same prompt. As the abilities of LLMs improve, we expect these approximation errors to reduce, as they are due to the language model's inconsistent imitation of a linear model.

This is an approximation of the LLM's in-context parameter distribution, giving new visibility into its predictive process.

### 5.5. Is the Language Model Consistent?

We compare these in-context priors with the AutoElicit priors. Table 1 suggests the LLM uses different priors during in-context learning compared to those we elicit, since they differ according to their energy statistic. This is supported by Appendix A.14, which presents histograms of the parameter values. Either the distribution we elicit is an unreliable representation of the LLM's knowledge, or it is unable to use this knowledge when making predictions in-context. As in-context priors are often centred around 0, we suspect it is the latter, and that prior elicitation more effectively makes use of the LLM's background knowledge.

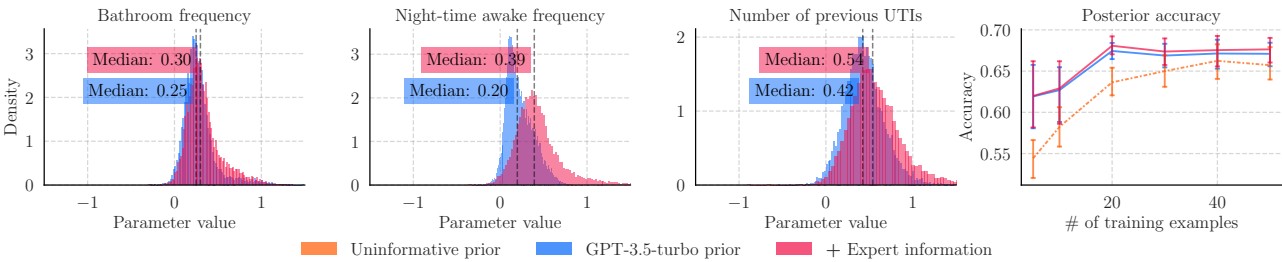

*Figure 3.* **UTI prior parameters with expert information.** The histograms in the three left-hand plots show the distribution of parameter values for the features that we provide expert information for in the task description. In all three cases, we state that the features are positively correlated with UTI risk. In the right-hand plot, we show the posterior accuracy of these distributions for different numbers of observed samples. The average and 95% confidence interval of the mean posterior accuracy over the 10 test splits are shown.

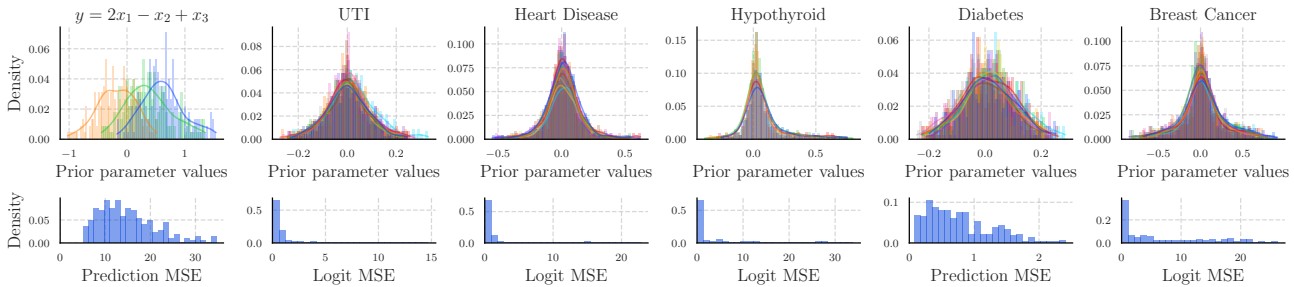

*Figure 4.* **Extraction of GPT-3.5-turbo's in-context prior.** The bottom row shows the MSE between the LLM's in-context predictions and the MLE model's predictions, whilst the top row shows the parameter distribution of these MLE models, with each colour representing a single feature. For ease of visualisation, we exclude values outside of the $(2.5\%, 97.5\%)$ percentiles and the bias term.

*Table 1.* **Approximated GPT-3.5-turbo in-context prior compared to the elicited prior.** The energy statistic between the LLM's approximate in-context prior and the LLM-elicited prior. Lower values indicate that the two distributions are more similar.

| UTI | Heart Disease | Hypothyroid | Diabetes | Breast Cancer |
|-----|---------------|-------------|----------|---------------|
| 0.39 | 0.18 | 0.15 | 0.38 | 0.31 |

### 5.6. Does In-context Learning use Bayesian Inference?

It is important that in-context learning performs Bayesian inference, so that new data updates the LLM's knowledge reliably. We empirically study this by comparing the approximated in-context posterior with the Monte Carlo (MC) posterior samples of the approximated in-context prior.

These two posterior samples are shown in Figure 6, where we focus on those datasets where in-context learning produced linear predictions, providing a low posterior approximation error. For the Diabetes task, samples from the approximated in-context posterior and MC sampling on the approximated in-context prior visually agree and produce a low energy statistic, suggesting the LLM is performing Bayesian inference in-context. However, for the Heart Disease task, the two posterior samples differ, showing that the LLM inconsistently applies the Bayes rule in-context.

Results of the other datasets and training splits are given in Appendix A.15. This suggests LLM-elicited priors are more reliable, particularly when access to the posterior distribution is important, such as for uncertainty estimation.

It is feasible that future LLMs can reliably perform Bayesian inference in-context, requiring a model selection strategy to decide between in-context learning and LLM-elicited priors.

### 5.7. Using the Bayes Factor for Model Selection

If in-context learning approximates Bayesian inference, we can calculate the Bayes factor between it and AutoElicit.

Figure 7 shows the marginal likelihood distribution for in-context learning, AutoElicit, and an uninformative prior, along with the Bayes factor (where reliably defined). Unlike in-context learning, AutoElicit priors are the better choice over an uninformative prior for all datasets tested. Whether AutoElicit or in-context learning provides better priors depends on the task. However, in only two of the five datasets tested, in-context learning reliably produced linear predictions, and in only one of those it approximated Bayesian inference. Considering this, the cost, and the ease of interpretability, AutoElicit is likely the better option.

This model selection strategy is designed to be more relevant as LLMs become more reliable and their costs reduce.

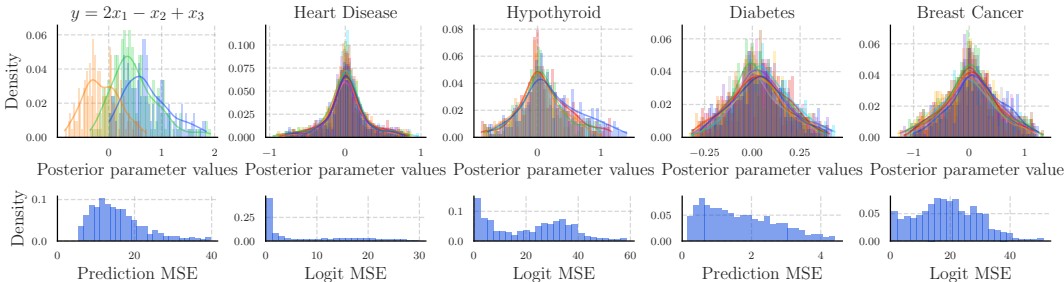

*Figure 5.* **Extraction of GPT-3.5-turbo's in-context posterior.** The bottom row shows the MSE between the LLM's in-context predictions and the MLE model's predictions, whilst the top row shows the parameter distribution of these MLE models, with each colour representing a single feature. For ease of visualisation, we exclude values outside of the $(2.5\%, 97.5\%)$ percentiles and the bias term.

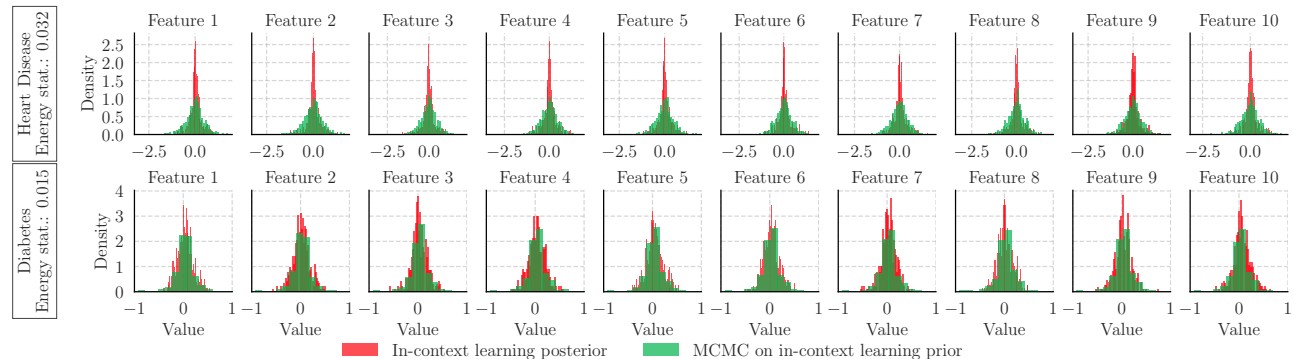

*Figure 6.* **The in-context posterior distribution.** We compare the posterior extracted from GPT-3.5-turbo's in-context predictions against the distribution that we sample using Monte Carlo methods on the extracted prior distribution. Both posterior distributions are based on the same 25 data point sample from the corresponding dataset. We draw the histogram separately in each dimension, but calculate the energy distance across all features combined. For the diabetes dataset, we have removed the long tails for improved visualisation.

# 6. Limitations and Future Research

This work explores how experts can guide AutoElicit priors with natural language, but studying other ways of incorporating experts into the framework would be interesting. For example, by fine-tuning LLMs on relevant literature.

As AutoElicit priors are a product of their training, they might contain unwanted bias. To avoid this, an approach would be to assign all features that are sensitive to unwanted bias, directly or indirectly, with an uninformative prior. Given an elicited prior and observed examples, the Bayes factor can then be used to choose between the elicited prior and an uninformative prior. Additionally, experts can steer LLMs away from unwanted bias through task descriptions.

Finally, it would be interesting to extend our results to neural networks. An immediate option is to build on the literature, asking the LLM to suggest data augmentation techniques for a task, effectively reducing the cost of data collection by extending the current dataset. Further, extending our results to non-linear models such as random forests or Gaussian processes would be valuable.

# 7. Conclusion

This work shows that, by asking a language model to play the role of a domain expert, we are able to elicit prior distributions for predictive tasks, potentially reducing the cost of data collection and improving the accuracy of our models when we have access to few labelled examples.

We find the priors elicited from GPT-3.5-turbo, GPT-4-turbo, and DeepSeek-R1-32B-Q4 achieve better accuracy or mean squared error with fewer data points than an uninformative prior for a variety of tasks. This improvement is most visible when few training examples are available, suggesting its use in environments where label collection is challenging.

Adding expert guidance to the task descriptions enables further improvements to predictive accuracy, as the LLM updates elicited distributions as we would expect. AutoElicit can therefore also be used to simplify the typical expert prior elicitation process by only requiring experts to provide insights in natural language. We find that prior elicitation with AutoElicit is an improved alternative to in-context learning in our setting, as we can inspect priors, confidently perform Bayesian inference, and often achieve reduced error.

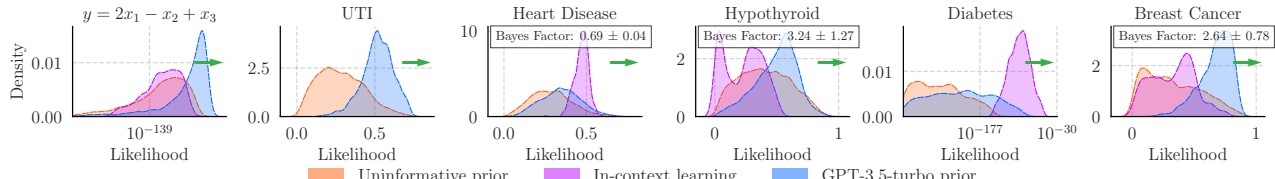

*Figure 7.* **The distribution of the prior marginal likelihood and the Bayes factor for GPT-3.5-turbo.** This shows the distribution of prior likelihoods, calculated using 25 data points from each dataset, over 5 test splits and 100 samples of prompts. The Bayes factor is calculated between prior elicitation and in-context learning, with a number greater than 1 favouring prior elicitation. For the regression tasks, we exclude values outside of the $(2.5\%, 97.5\%)$ percentiles in the histograms and assume the priors form a single Gaussian.

On our private dataset, we find a large and quantifiable performance improvement when predicting cases of UTI. Within our study on dementia care, the resulting model created with LLM-elicited priors can provide better predictions of adverse health conditions, reliable uncertainty estimates, and remain interpretable and trusted by clinicians.

## Impact Statement

This paper presents work that aims to make expert prior elicitation more accessible and suggests a new way for language models to be used in settings where high levels of trust in predictive models are required. We see many potential societal consequences of our work, none of which we feel must be specifically highlighted here.

## Author Contributions

**AC:** Original idea conception, software, analysis of results, writing, editing, reviewing, funding acquisition. **RGK:** Original idea conception, supervision, analysis of results, editing, reviewing, funding acquisition. **PB:** Supervision, analysis of results, reviewing, funding acquisition.

## Acknowledgments

This study is funded by the UK Dementia Research Institute (UKDRI) Care Research and Technology Centre funded by the Medical Research Council (MRC), Alzheimer's Research UK, and Alzheimer's Society (grant number: UK-DRI–7002). Further funding was provided by the UKRI Engineering and Physical Sciences Research Council (EPSRC) PROTECT Project (grant number: EP/W031892/1) and the Royal Academy of Engineering, UK. Infrastructure support for this research was provided by the NIHR Imperial Biomedical Research Centre (BRC) and the UKRI Medical Research Council (MRC). Additionally, funding was provided by the Turing scheme, UK, and resources used in preparing this research were provided, in part, by the Province of Ontario, the Government of Canada through CIFAR, and companies sponsoring the Vector Institute. Moreover, RGK is supported by a Canada CIFAR AI Chair and a Canada Research Chair Tier II in Computational Medicine. The funders were not involved in the study design, data collection, data analysis, or manuscript writing.

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

# A. Appendix

## A.1. Availability of Code

We provide the code to generate all figures and results[2], and functions to use our method on new datasets[3], on GitHub.

This open-source code makes it simple to perform expert prior elicitation using LLMs:

```python
# import the elicitation function
from llm_elicited_priors.gpt import (
    get_llm_elicitation_for_dataset
)

# elicit priors for a dataset
get_llm_elicitation_for_dataset(
    # the language model client
    # requires a .get_result(messages)
    client=llm_client,
    # the task descriptions
    # to iterate over
    system_roles=system_roles,
    user_roles=user_roles,
    # the dataset feature names
    feature_names=data.feature_names,
)
```

We provide classes that wrap common LLMs (Appendix A.16), whilst making it easy to adapt to new models.

This code is implemented in python (v3.11.9), using pip (v24.0). For our experiments, we use the following packages:

___

- blackjax (v1.2.3) (Cabezas et al., 2024): For Monte Carlo sampling on the in-context prior samples.

- jax (v0.4.31) (Bradbury et al., 2018): For calculating some metrics and use with blackjax.

- numpy (v1.26.4) (Harris et al., 2020): For manipulation of arrays and computing mathematical equations.

- ollama (v0.4.7): For loading the DeepSeek model.

- openai (v1.51.2): For communicating with the OpenAI API.

- pandas (v2.2.2) (Wes McKinney, 2010): For manipulating the datasets and results.

- pymc (v5.17.0) (Abril-Pla et al., 2023): For Monte Carlo sampling on the elicited priors.

- scikit-learn (v1.5.1) (Pedregosa et al., 2011): For loading and preprocessing the data.

- transformers (v4.46.0): For loading and using Llama models.

A requirements file is provided within the supplementary code, which also includes some other utility packages required to run the scripts and load the data and results.

## A.2. Further Details on the Related Work

In this section, we outline in more detail the prior work in our area of research.

### A.2.1. USING LLMS TO ELICIT EXPERT PRIORS

Choi et al. (2022) explores the use of an LLM to perform feature selection, causal discovery, and to modify the reward function in a reinforcement learning task. In the first case, they observe that when listing features from two datasets, GPT-3 can correctly identify the features that correspond to a single specified predictive task. Additionally, when applied to causal discovery, the language model is often able to predict the directionality of causal relationships in the Tüebingen Cause-Effect Pairs dataset (Mooij et al., 2016). This inspires us to explore the possibility of using LLMs to elicit prior parameter distributions over linear models.

Gouk & Gao (2024) use LLMs to generate synthetic data to elicit priors on public datasets. A logistic regression model is then pre-trained using this data, defining a prior. However, this work does not discuss the memorisation of public datasets by LLMs, which is shown to be substantial (Bordt et al., 2024), supported by the results in Appendix A.8. In Appendix A.9.5, we compare the posterior performance of AutoElicit with the method proposed in Gouk & Gao (2024)

and show that on private data, AutoElicit provides more informative priors.

Further, Selby et al. (2025) elicit beta distribution parameters to define a prior for predictive tasks where results from human experts are available. The priors from four LLMs were visually compared to those elicited from human experts using the SHELF (Oakley & O'Hagan, 2019; Gosling, 2018) method, demonstrating significant differences between experts and LLMs. However, by considering prior predictive distributions, they show that GPT-4 (OpenAI et al., 2024) can provide parameter distributions that allow for zero-shot performance comparable to models trained on tens of data points. These priors are over the targets directly, as with in-context learning, and are not updated using data. In our work, we use the LLM to elicit multiple Gaussian distributions, which are combined to create a mixture model for a more general distribution over the parameters of a linear predictive model. Our priors can then be updated using observed data. Further, we explore the collaboration of LLMs and human experts to provide more informative prior distributions.

In addition, Requeima et al. (2024) prompt language models with task descriptions and collected data points to generate posterior predictive distributions for regression tasks, which they find outperform Gaussian processes in some tasks. This is done by repeatedly sampling a language model at a new feature value, given the previous input and output observations, and a given text description that contains prior information from a human expert. In our work, we explore whether repeatedly sampling language models can also produce general prior distributions for predictive models.

Furthermore, Zhu & Griffiths (2024) describe the use of iterated learning to elicit prior distributions from language models for causal learning, proportion estimation, and predicting everyday quantities. The authors employ a Markov chain Monte Carlo method to access an LLM's prior knowledge by performing successive inference using previous model outputs that tend towards a limiting distribution. In doing this, the authors assume that in-context learning is a form of Bayesian inference. However, in their work Zhu & Griffiths (2024) do not consider how prior elicitation from LLMs can be used for predictive models or test whether an LLM is performing Bayesian inference, both of which we study in this work.

Other works explore using LLMs for defining entire modelling pipelines (Tornede et al., 2023; Zhang et al., 2023; Li et al., 2024). In particular, Li et al. (2024) questioned whether Box's Loop (Box & Hunter, 1962) could be improved upon by using LLMs. In this work, they were used as model builders and critics to automate the model design and iteration steps. The authors find that this process can produce models that perform well in various modelling tasks

and are comparable to those produced by domain experts. This demonstrates that LLMs have a good understanding of probabilistic modelling, suggesting that they might also be able to provide priors over predictive models directly.

Li et al. (2023) discusses using LLMs to produce prior distributions for perception and control tasks. This work focuses on vision and robotics and elicits common-sense priors from LLMs to bias perception and action models towards realistic world states. The authors find that for vision tasks, this improves predictions on rare, out-of-distribution, and structurally novel inputs. Our work aims to extend this by evaluating whether LLMs can also provide expert information about parameters in predictive models.

### A.2.2. IN-CONTEXT LEARNING

Learning from demonstrations in-context has proven to be effective for several natural language tasks, demonstrating the surprising ability of language models to perform outside of their intended use (Brown et al., 2020; Dong et al., 2024; Hegselmann et al., 2023; Chowdhery et al., 2023; Touvron et al., 2023). In Brown et al. (2020), the authors study GPT-3's strong ability on natural language tasks after seeing a few examples, one example, or none at all. This work inspired others to study the extent to which language models can learn from demonstrations given as text alongside a task description. Further, Xie et al. (2021) study the mechanics of in-context learning and describes it from a Bayesian Inference perspective. They find that in-context learning succeeds when language models can infer a shared concept across individual examples in a prompt. Min et al. (2022) argue that in-context learning allows the language model to observe the label space, the distribution of input text, and the task format, as they find that swapping the real labels in demonstrations for random labels does not significantly reduce accuracy. Most of the existing work focuses on natural language tasks (Dong et al., 2024), leaving numerical problems less explored and a focus of our work.

In Hollmann et al. (2022), the authors train a transformer model to solve synthetically generated classification tasks from a prior over tabular datasets. Given a new task and demonstrations, this transformer performs in-context learning to make predictions using inference steps only. Furthermore, Akyürek et al. (2022) and Von Oswald et al. (2023) discuss the types of learning algorithms that language models (and more generally, transformers) can implement in-context. Focusing on linear models, Akyürek et al. (2022) show that, by construction, transformers are capable of performing gradient descent and closed-form ridge regression. Akyürek et al. (2022) also demonstrate that trained in-context learners closely match predictors trained through gradient descent. Further, Von Oswald et al. (2023) show that transformers, by incorporating multilayer perceptrons

into their architectures, can also solve non-linear regression tasks, and Dalal & Misra (2024) propose a Bayesian framework to reason about a language model's internal state and suggest that in-context learning is consistent with Bayesian learning. Wang et al. (2023), by viewing LLMs as latent variable models, suggest that in-context learning can reach the Bayes optimal predictor with a finite number of demonstrations, which can be chosen from a dataset using their framework. However, the authors Falck et al. (2024) provide evidence that LLMs violate the martingale property, which they show is a requirement of a Bayesian learning system for exchangeable data. Through this, they argue that the in-context learning behaviour LLMs exhibit is, in fact, not Bayesian.

In contrast to the literature, in our work we empirically test, through the approximation of the internal model used by a language model for its in-context predictions, whether for a given task the language model is performing Bayesian inference before and after seeing demonstrations.

### A.3. Formal Details on Extracting the Internal Prior and Posterior

In this section, we describe the method presented in Section 4.2 in more detail.

Given a language model $M$ and task $T$ that provides in-context predictions, if we can construct a function that allows us to sample a single parameterisation $\varphi \sim p_{M,T}(\varphi)$, then we can estimate $p_{M,T}(\varphi)$ using a sampling distribution. Following Murphy (2022), we define an estimator:

$$\varphi_{\text{MLE}} = \text{MLE}_{M,T}(\hat{\mathcal{D}}) = \arg\max_{\tilde{\varphi}} p_{M,T}(\hat{\mathcal{D}}|\tilde{\varphi}) \quad (5)$$

This corresponds to the maximum likelihood estimate (MLE) of $\varphi$ over some dataset $\hat{\mathcal{D}}$ given the task $T$. Now consider that $\hat{\mathcal{D}}$ is generated by a given $\varphi$ sampled from $\varphi \sim p_{M,T}(\varphi)$ and $X$. By sampling $x \sim \mathcal{U}[X]$ from a uniform distribution over $X$, we can create $S$ different datasets of the form:

$$\hat{\mathcal{D}}^{(s)} = \{y_n \sim p_{M,T}(y_n|x_n, \varphi) : n = 1:N, \; x \sim \mathcal{U}[X]\} \quad (6)$$

This provides us with the sampling distribution:

$$\varphi_{\text{MLE}}^{(s)} \sim \text{MLE}_{M,T}(\hat{\mathcal{D}}^{(s)}) = \arg\max_{\tilde{\varphi}} p_{M,T}(\hat{\mathcal{D}}^{(s)}|\tilde{\varphi})$$

With this, we can generate a sampling distribution using $\{\varphi_{\text{MLE}}^{(s)}\}_{s=0}^{s=S}$ that approximates $p_{M,T}(\varphi)$ as $S \to \infty$. Furthermore, by further incorporating the training dataset $\mathcal{D}$:

$$\hat{\mathcal{D}}^{(s)} = \{y_n \sim p_{M,T}(y_n|x_n, \varphi, \mathcal{D}) : n = 1:N, \; x \sim \mathcal{U}[X]\}$$

We get the sampling distribution:

$$\varphi_{\text{MLE}}^{(s)}|\mathcal{D} \sim \text{MLE}_{M,T}(\hat{\mathcal{D}}^{(s)}|\mathcal{D}) = \arg\max_{\tilde{\varphi}} p_{M,T}(\hat{\mathcal{D}}^{(s)}|\tilde{\varphi}, \mathcal{D})$$

Allowing us to estimate $p_{M,T}(\varphi|\mathcal{D})$.

### A.4. Does the LLM use the Model Class we Specify?

As part of comparing AutoElicit with in-context learning, we use maximum likelihood sampling to extract the internal parameter distributions (discussed in Section 5.4), which we use to test whether the LLM is: (1) providing a similar prior distribution through elicitation to the one it is using in-context, and (2) empirically test whether the LLM is approximating Bayesian inference. However, when extracting the internal parameters of the in-context model, we assume that the LLM is applying the model class that we specify in the task description.

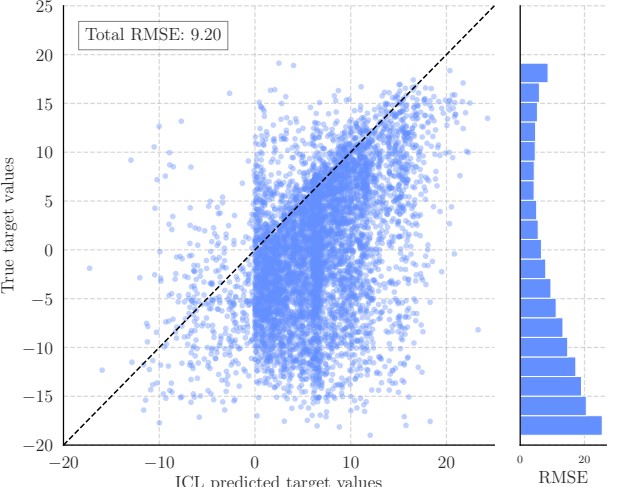

*Figure 8.* **GPT-3.5-turbo in-context predictions for the task:** $y = 2x_1 - x_2 + x_3$ **compared to the true values.** Here, feature values are generated using $X \sim \mathcal{U}(-5, 5)$ and the targets are calculated using $y = 2x_1 - x_2 + x_3$, which is specified fully in the in-context prompt. The language model is then asked to make predictions on $X$. This shows the true targets $y$ against those predicted by the language model, as well as the binned root mean squared error.

However, we find that for some datasets, the language model does not use the same model in context as the one we specify. For the synthetic data in particular, we specify the relationship as $y = 2x_1 - x_2 + x_3$. Here, we can measure whether GPT-3.5-turbo is using the specified model in-context to predict the target by calculating the error between the true $y$ values and those returned by the language model's in-context predictions. This is done by measuring the mean squared error between the $y_n$ given in Equation 6 and the true $\hat{y}_n = 2x_{n,1} - x_{n,2} + x_{n,3}$ calculated using the $x_n$ in Equation 6.

In Figure 8 we show the true $\hat{y}_n$ against the in-context predictions $y_n$. Interestingly, even given the simple linear equation we ask the language model to mimic, we find that it struggles to provide good predictions in-context. This suggests that the language model is not always using the model class for

in-context predictions that we specify in the task description, indicating that, in its current state, in-context learning might be inadequate for predictive tasks with primarily numerical features. This figure also shows that the language model was hesitant to predict negative values and preferred positive predictions – a strange behaviour that clearly restricts its use as an emulator of a linear model.

However, for all other prior extraction tasks presented in Section 5.4, we find that the language model uses a linear model in-context, demonstrated by the low MSE loss values between the predictions of our MLE samples and the true in-context predictions, shown in Figure 4. This is often not the case when the language model is presented with demonstration samples, as in the posterior extraction presented in Figure 5, in which the language model uses a linear combination of features to predict the targets for the Heart Disease and Diabetes tasks only.

## A.5. Energy Distance and Statistic

The energy statistic (Székely & Rizzo, 2013), which is based on the energy distance metric (Székely & Rizzo, 2012), is a method to measure the similarity of two probability distributions. Importantly, it allows us to test whether two $N$-dimensional probability distributions are equal.

The energy distance $D(X, Y)$, with $X$ and $Y$ as independent random vectors $\in \mathbb{R}^n$ is defined as:

$$D(X, Y)^2 = 2\mathbb{E}[M_{XY}] - \mathbb{E}[M_{XX}] - \mathbb{E}[M_{YY}]$$

Here, $M_{XY\,i,j} = ||x_i - y_j||_2$ is the pairwise distance matrix.

From this, we construct the energy statistic as follows:

$$H(X, Y) = \frac{D(X, Y)^2}{2\mathbb{E}[M_{XY}]} \qquad (7)$$

This statistic takes values between 0 and 1 and tests whether two statistical distributions are equal.

We provide an implementation of this distance metric and test statistic within our supplementary code (Appendix A.1).

## A.6. The Bayes Factor

Given two models $\alpha_0$ and $\alpha_1$ with parameterisations $\theta_0$ and $\theta_1$, the Bayes factor calculated over a dataset $\mathcal{D}$ is:

$$\text{BF}(\alpha_0, \alpha_1; \mathcal{D}) = \frac{p(D|\alpha_0)}{p(D|\alpha_1)} = \frac{\int_\Theta p(\theta_0|\alpha_0)p(D|\theta_0, \alpha_0)d\theta_0}{\int_\Theta p(\theta_1|\alpha_1)p(D|\theta_1, \alpha_1)d\theta_1} \qquad (8)$$

In our case, $\alpha_0$ refers to prior elicitation, with the prior determining the parameterisation $\theta$ of a predictive model, and $\alpha_1$ referring to in-context learning. We can approximate the Bayes factor using the means from samples of the likelihood of $D$.

We chose the Bayes factor over the Bayesian Information Criterion (BIC) because BIC uses the number of model parameters as part of the model selection. When comparing prior elicitation with in-context learning, BIC would almost always favour prior elicitation because the number of parameters required to make a prediction using in-context learning is of the order $10^9$.

## A.7. Further Experimental Details

In all experiments, we use OpenAI's GPT-3.5-turbo (specifically GPT-3.5-turbo-0125), GPT-4-turbo (GPT-4-turbo-2024-04-09)[4], and DeepSeek-R1-32B-Q4[5] with a temperature of 0.1 and all other settings as default.

### A.7.1. DATASET DETAILS

In this work, we evaluate our method on 6 datasets; 1 of which is synthetic, 1 is privately collected, and 4 are publicly accessible. This allows us to provide both reproducible experiments and test the language model's ability to suggest a good prior on datasets not seen during training.

- $y = 2x_1 - x_2 + x_3$: To evaluate language models as tools for prior elicitation, we construct a synthetic dataset where we have control over the parameters of the dataset. Because of this, we are able to provide the language model with a complete description of the task. This dataset contains 3 features sampled from a normal distribution (mean $= 0$ and standard deviation $= 1$) and targets calculated using the equation $y = 2x_1 - x_2 + x_3 + \epsilon$, where $\epsilon \sim \mathcal{N}(0, 0.05)$. When training and testing were done on these samples, no preprocessing was performed.

- A private dataset collected as part of our study on Dementia care[6]: This dataset contains 10 features engineered from measurements of in-home activity and physiology. The raw features were measured using passive infrared (PIR) sensors located around the homes of people living with dementia, and a sleep mat positioned in the participants' beds. The targets correspond to days with a clinical labelling of positive or negative urinary tract infection (UTI) which were calculated through the analysis of a provided urine sample in consultation with a healthcare professional (Capstick et al., 2024). For this analysis, we used a subset of the data containing 78 people and 371 labelled days of UTI, with 270 negatives and 101 positives. When training

---

[4] https://platform.openai.com/docs/api-reference/chat
[5] https://ollama.com/library/deepseek-r1:32b
[6] https://www.imperial.ac.uk/uk-dri-care-research-technology/

and testing on this dataset, we centred and scaled the training data and used the same statistics to normalise the test data.

- The Heart Disease dataset (Detrano et al., 1989; Andras Janosi, 1989): This dataset contains 10 physiological features and targets with a positive or negative diagnosis of heart disease for 299 people. Within this dataset, we have 161 examples of people without heart disease and 138 examples of people with heart disease, totalling 299 data points. When training and testing on this dataset, we centred and scaled the training data and used the same statistics to normalise the test data, except where features are categorical. In the categorical case, we perform no scaling.

- The Hypothyroid dataset (Quinlan, 1986; Turney, 1994): This dataset, processed from the original, consists of 150 examples of patients' thyroid-related blood levels and whether they have a hypothyroidism diagnosis or not. This dataset contains 4 features, with 75 examples of a patient with hypothyroidism and 75 examples of a patient without hypothyroidism. When training and testing on this dataset, we centred and scaled the training data and used the same statistics to normalise the test data.

- The Diabetes dataset (Efron et al., 2004): After preprocessing, this dataset contains physiological measurements and demographic information from 442 people with diabetes. Here, the task is to predict a quantitative measure of their diabetes disease progression one year after this baseline. This dataset consisted of 10 features, and the targets were regression values with a maximum value of 10.0, a minimum value of 0.0, and a mean value of 3.96.

- The Breast Cancer dataset (Wolberg et al., 1995): This dataset is made up of 569 data points with 30 features computed from an image of a breast mass that are labelled as either malignant or benign. In total, this dataset contains 357 examples of benign tumours and 212 examples of malignant tumours. When training and testing on this dataset, we centred and scaled the training data and used the same statistics to normalise the test data.

The UTI study was performed in collaboration with Imperial College London and Surrey and Borders Partnership NHS Trust. Participants were recruited from the following: (1) health and social care partners within the primary care network and community NHS trusts, (2) urgent and acute care services within the NHS, (3) social services who oversee sheltered and extra care sheltered housing schemes, (4) NHS

Community Mental Health Teams for older adults (CMHT-OP), and (5) specialist memory services at Surrey and Borders Partnership NHS Foundation Trust. All participants provided written informed consent. Capacity to consent was assessed according to Good Clinical Practice, as detailed in the Research Governance Framework for Health and Social Care (Department of Health 2005) and the Mental Capacity Act 2005. Participants were provided with a Participant Information Sheet (PIS) that includes information on how the study uses their personal data collected in accordance with the GDPR requirements. If the participant was deemed to lack capacity, a personal or professional consultee was sought to provide written consent to the study. Additionally, the capacity of both the participant and study partner is assessed at each research visit. Research staff conducting the assessment have completed the NIHR GCP training and Valid Informed Consent training. If a participant is deemed to lack capacity but is willing to take part in the research, a personal consultee is sought in the first instance to sign a declaration of consent. If no personal consultee can be found, a professional consultee, such as a key worker, is sought. This process is included in the study protocol, and ethical panel approval is obtained.

Where datasets are publicly available, we perform minimal processing for ease of reproducibility. In some cases, we replace the feature names with ones that are more descriptive (taken from the dataset descriptions also available publicly) to aid the language model's comprehension of the data. All of these changes to the raw data are available within the supplementary code, which will automatically download and process the data when loading.

### A.7.2. LANGUAGE MODEL DETAILS

Within the main experiments, we evaluate three language models:

- GPT-3.5-turbo[7] is one of OpenAI's least expensive models and is described as a model for simple tasks.

- GPT-4-turbo[7] is one of OpenAI's more sophisticated models.

- DeepSeek-R1-32B-Q4[8], a 32 billion parameter model from DeepSeek based on Qwen 32B and distilled from DeepSeek-R1.

These represent three distinct types of language models. GPT-3.5-turbo is a cost-effective option for those who are able to use OpenAI's API and removes the need to run the language model locally. Similarly, GPT-4-turbo is hosted by

---

[7]https://platform.openai.com/docs/models
[8]https://ollama.com/library/deepseek-r1:32b

OpenAI, however, it is a considerably more expensive option. In contrast, DeepSeek-R1-32B-Q4 is an open-source model that can be run locally, given the compute requirement is met, allowing priors to be elicited in a secure environment and without API costs.

### A.7.3. EVALUATING LLM-ELICITED PRIORS

To evaluate the usefulness of the AutoElicit priors (presented in Section 4.1), we started by comparing their posterior performance to an uninformative prior. To do this, we chose the predictive tasks described in Section 5.1: $y = 2x_1 - x_2 + x_3$, Urinary Tract Infection (UTI), Heart Disease, Hypothyroidism, Diabetes, and Breast Cancer and calculated the posterior accuracy or mean squared error on a test set after observing varied numbers of examples. For each of the datasets, we selected $50\%$ of the data points to form the test set and used the remaining data points to form a training set. This is done 10 times to produce 10 folds of random training and testing splits.

For the UTI task, we split the dataset so that each participant's data is only in either the testing or training set, as some participants produced multiple examples. For the other classification tasks, the splits are stratified using the targets. To evaluate the posterior performance after observing a given number of data points, we randomly select that number of examples from the training set to train on and then test on the same test set as before.

After performing prior elicitation (Section 5.2), for each of the tasks, we have 100 Gaussian priors for each feature in the dataset. We then build a mixture of the priors using Equation 2, where we weight the Gaussian distribution in each dimension and each task description separately.

We then construct a logistic or linear regression model, where we set the prior distribution over the bias term as a mixture of 100 distributions of $\theta_0 \sim \mathcal{N}(0, 1)$. In the case of linear regression, we also use a noise term $\epsilon \sim \text{Half-Cauchy}(\beta = 1)$, with a Half-Cauchy prior. In the uninformative case, we set $\theta \sim \mathcal{N}(0, 1)$ for each feature.

We then use the No-U-Turn sampler (Hoffman et al., 2014) to sample from the posterior distribution over the linear model parameters, with 5 chains and 5000 samples per chain. In total, for each dataset, for each training and testing split, we have 25,000 posterior samples over the posterior.

To produce the results in Figure 9, we then sample the posterior predictive distribution over our test set and calculate the accuracy or mean squared error for each posterior parameter sample. To produce Figures 2 and 14, we also calculate the mean accuracy and the mean squared error for each of the test splits before plotting the mean and the $95\%$ confidence interval.

As an example, for the synthetic task, one of the task descriptions is as follows:

- **System role:** You're a linear regression predictor, estimating the target based on some input features. The known relationship is: 'target' = 2 * 'feature 0' - 1 * 'feature 1' + 1 * 'feature 2'. Use this to predict the target value.

- **User role:** I am a data scientist working with a dataset for a prediction task using feature values to predict a target. I would like to apply your model to predict the target for my samples. My dataset consists of these features: ['feature 0', 'feature 1', 'feature 2']. All the values have been standardised using z-scores. The known relationship is: 'target' = 2 * 'feature 0' - 1 * 'feature 1' + 1 * 'feature 2'. Based on the correlation between each feature and target, whether positive or negative, please guess the mean and standard deviation for a normal distribution prior for each feature. I need this for creating a linear regression model to predict the target. Provide your response as a JSON object with the feature names as keys, each containing a nested dictionary for the mean and standard deviation. A positive mean suggests positive correlation with the outcome, negative for negative correlation, and a smaller standard deviation indicates higher confidence. Only respond with JSON.

### A.7.4. EXTRACTING THE IN-CONTEXT MODEL'S HIDDEN PRIOR AND POSTERIOR

To further study in-context learning, we used maximum likelihood sampling to extract approximate samples of the internal model's prior and posterior using the method described in Section 4.2. For each of the tasks studied, we randomly generate 25 feature values from $X \sim \mathcal{U}[-5, 5]$ 5 times and ask the language model to predict their regression label or positive classification probability based on the task description given and with the specified model class (in our case a linear or logistic regressor). This is done for each of the 100 task descriptions that we have for the dataset. We then fit a linear regressor to the predicted regression labels or the classification logits[9] of the predicted probabilities, using maximum likelihood estimation. The parameters of this linear model are then taken as a single sample of the approximated in-context predictive model.

For each of these linear model samples, we calculate the regression labels or the classification logits using the same data given to the language model for in-context predictions. This allows us to calculate an error in our estimation of the

---

[9]Calculated using the inverse logistic function: logits $= \ln p/(1 - p)$

in-context model, which depends on how well the in-context learner is approximating a linear model.

In total, for each task description, we collect 5 samples of the in-context model parameters. Once this is repeated for all of the 100 task descriptions, we have 500 samples of the in-context model.

We can then measure, as in Section 5.5, whether the parameter samples are similar to those sampled from the elicited distributions. This is done by sampling 10,000 parameter values from the elicited distributions and calculating the energy statistic (Appendix A.5) between this sample and the extracted in-context sample, and through visual inspection. By doing this, we test whether the language model is using the same distribution we elicit from it as the one it uses in-context. The results of this are presented in Table 1.

When extracting the posterior distribution over the in-context learner, we repeat the above process 5 times, where in each repeat we provide the language model with 25 demonstration points from the corresponding dataset. This allows us to empirically question whether the language model is approximating Bayesian inference when it updates the in-context prior distribution with the observed data points.

To do this, we start by fitting a Gaussian kernel density estimator (with a bandwidth factor of 0.25) to our in-context model prior parameter samples. We then use the No-U-Turn sampler (Hoffman et al., 2014) with 1000 adaption steps and the same demonstration examples given during in-context learning to sample posterior parameters. Here, we use 100 chains of 10,000 samples to build an approximate posterior distribution.

We can then use the energy statistic to measure the difference between the extracted in-context posterior distribution and the Monte Carlo approximation of the posterior based on the extracted in-context prior distribution. The values presented in Table 4 show the result of this experiment.

### A.7.5. CALCULATING THE BAYES FACTOR

As part of our solution to decide whether to use prior elicitation or in-context learning for a given dataset, we calculate the Bayes factor (presented in Section 4.3).

For a single dataset, we start by randomly selecting 25 data points, 5 times, and consider the prior predictive distribution. For prior elicitation, this involves sampling the prior predictive distribution (drawing 500 samples for each split), and for in-context learning, this involves prompting a language model for a prediction for each of the 100 task descriptions (producing 100 samples for each of the splits).

For both sets of prior predictive samples, we calculate their log-likelihood using either cross-entropy loss or mean squared error, depending on whether the task is classification or regression. For each dataset, for each test split, and for each method, we calculate the mean of this log-likelihood over the samples. With these values, we measure the difference between the value for the elicited prior predictions and the in-context predictions to obtain the log Bayes factor.

We then report the mean and standard deviation of the Bayes factor over the splits in Figure 7, which helps us to decide which method is most appropriate for a dataset; assuming that in-context learning approximates Bayesian inference.

### A.7.6. GIVEN A NEW DATASET

To apply the prior elicitation techniques to a new dataset, we recommend the following steps:

1. First, create a description of the dataset along with a description of the language model's role. As part of this, provide descriptive features and target names. It will be helpful to split this task description into what the language model is expected to do (system role) and the task that the user is giving it (user role). Please see our collection of task descriptions in the supplementary code or the example given in Appendix A.7.3.

2. Ask a language model to rephrase the user role and system role many times. In our experiments, we asked the language model to rephrase both descriptions 10 times for each.

3. Take the product of the system roles and user roles and provide each to the language model. For each of the unique system and user role combinations, record the mean and standard deviation of the Gaussian prior elicited from the language model.

4. Use these individual Gaussian priors to build a mixture of Gaussians as a prior distribution for a linear model.

5. Using the data points you have available from the predictive task, sample from the posterior predictive distribution to predict on unseen data.

Functions for doing this are provided in the supplementary code.

To measure whether prior elicitation or in-context learning is better for a given task (assuming the language model is approximating Bayesian inference):

1. Firstly, sample the prior predictive distribution, with the elicited priors found above, on the data points already collected.

2. Then, to perform in-context learning, use a similar task description to the above but with a different system

and user role that specifies the language model should make predictions instead of providing prior distributions. With this new set of descriptions, again ask a language model to rephrase them.

3. To sample the prior predictive distribution using the product of these descriptions, ask a language model to make predictions on the data directly.

4. Measure the Bayes factor between the likelihood of the prior predictive distribution using prior elicitation and the in-context model on the already collected data.

5. Using the Bayes factor and by factoring in resource cost, decide whether in-context learning or prior elicitation is more appropriate.

To empirically check whether the language model is approximating Bayesian inference to produce its in-context predictions or to test whether it is being faithful to the priors we elicit, we can follow the procedure below:

1. Using the same task descriptions as above, provide the language model with randomly generated feature values (we used 25 samples from $X \sim \mathcal{U}[-5, 5]$) for each task description and ask it to make predictions in-context.

2. By additionally providing the seen data as demonstrations to the language model, again ask the language model to make predictions on randomly generated feature values for each task description.

3. With randomly generated feature values and in-context predictions, use maximum likelihood estimation to estimate the underlying linear model for each task description. For the classification case, this is done more easily if the language model is asked to provide the probability of a positive prediction rather than the prediction itself.

4. The resulting linear models approximated from the in-context predictions can now be used to approximate the prior and posterior distribution of the in-context model.

Using the above, we can now (1) empirically test whether the language model is being faithful to the prior distributions we elicit and (2) empirically test whether the language model is approximating Bayesian inference in-context.

For the former, we can use a measure of the difference in samples (in our case, the energy distance, Appendix A.5) to calculate the difference between the in-context model's prior distribution and the elicited prior distribution. This will inform us of whether the language model is being truthful when it provides us with a prior on the parameters of

a linear model, or whether the language model uses that prior knowledge when making predictions in-context. If the prior distribution elicited differs from the one that the language model is using in-context, then calculating the Bayes factor could be important to ensure that we select the most appropriate predictive model for the given task. If it is not clear whether the language model is approximating Bayesian inference in-context, we should then study the latter to ensure the Bayes factor is a suitable model selection tool.

For this, we can follow the procedure below:

1. Use a kernel density estimator (KDE) to approximate a smoother version of the prior distribution over the in-context model. For our experiments, we used a Gaussian kernel density estimator with a bandwidth factor of 0.25.

2. Using the KDE and Monte Carlo methods, given the collected data, we sample a posterior distribution from the in-context learner's prior.

3. Use a method to calculate the difference between two samples of a distribution to understand the differences between the posterior samples calculated through Monte Carlo sampling of the in-context prior and the samples of the in-context posterior.

This last procedure might indicate that our in-context learner is not approximating Bayesian inference and is not suitable for a given task, since it could be important to us that our predictive model is updating its prior knowledge in a reliable way.

### A.8. Dataset Memorisation

To better contextualise the use of in-context learning in the tested datasets, we will use the method proposed in Bordt et al. (2024) to quantify the extent to which GPT-3.5-turbo already knows information about the datasets.

We start with the header completion test, which checks whether the language model has memorised the first few rows of the dataset. Here, the column names along with a given number of the first rows are given to the language model as a string, with the final of these rows truncated. The language model is then asked to generate the next 500 tokens.

We then use the Levenshtein distance (Levenshtein, 1965; 1966), a string matching metric, to calculate the agreement between the true header completion and the one provided by the language model. The Levenshtein distance counts the number of single-character changes required to transform one string into another, only allowing for insertions,

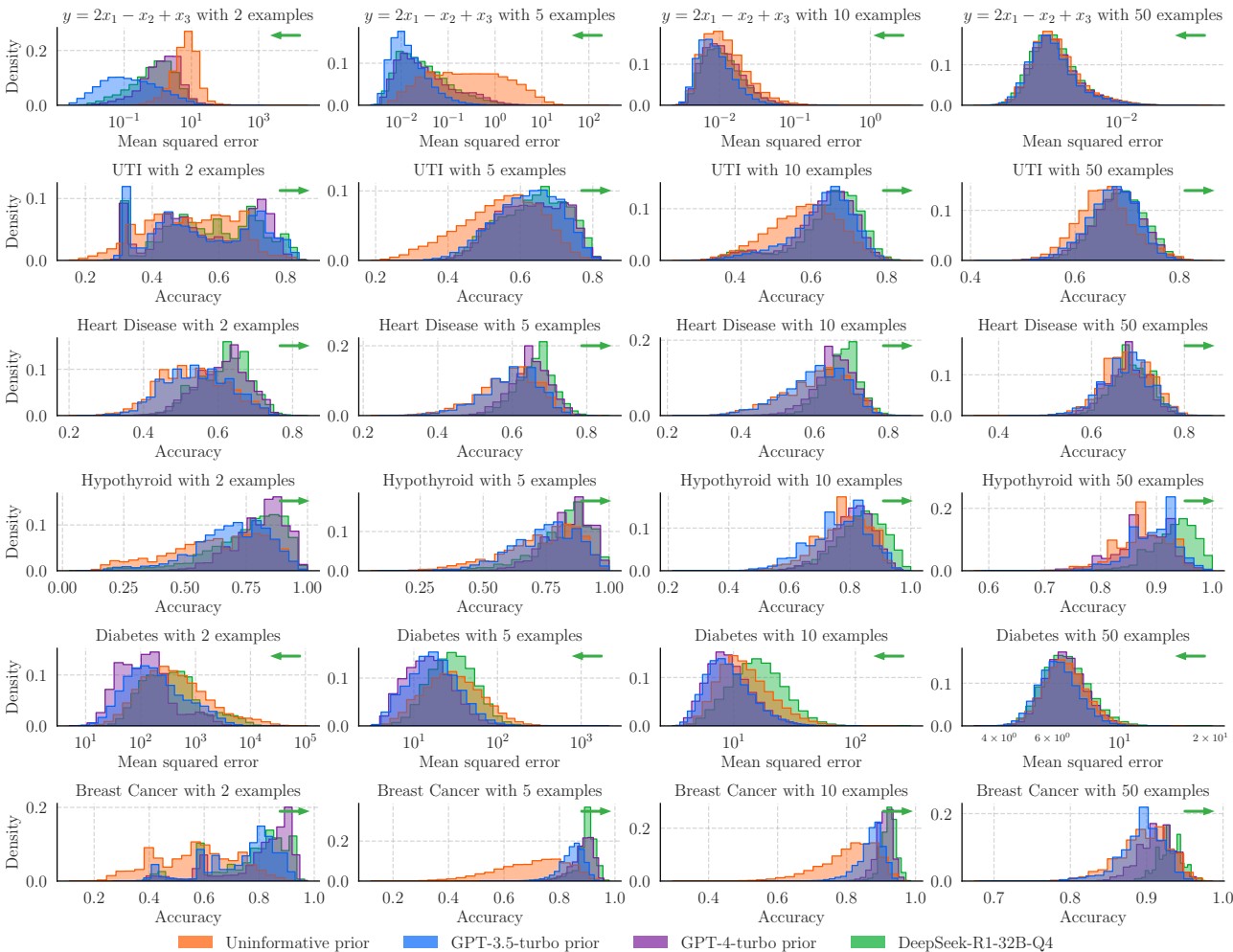

*Figure 9.* **Test accuracy of the posterior distribution over the model weights of a logistic and linear regression model for varied numbers of training data points.** We evaluate the accuracy and mean squared error of different prior distributions over the weights of a logistic and linear regression model. Each of the Gaussians in the mixture is extracted from a language model and is tested against an uninformative prior $\mathcal{N}(\theta|0, 1)$. These results are calculated on a test set of the datasets after each model is trained on the given number of data points. These values are the result of 5000 sampled parameters and 10 random splits of the training and testing data. The green arrows point in the direction of improvement in the metric.

deletions, and substitutions. The value can take a minimum value of 0, suggesting that the strings are equal, and a maximum value of the longest length of the two strings.

To make the Levenshtein distance comparable across datasets, we calculate its normalised value by dividing the Levenshtein distance by the length of either the true string or the language model's completion, whichever is longer. Because of this, the normalised Levenshtein distance reported is not formally a metric as it does not follow the triangle inequality. However, it allows us to compare the header completions across datasets more easily:

$$\hat{L}(s_1, s_2) = \frac{\text{Levenshtein}(s_1, s_2)}{\min\{|s_1|, |s_2|\}}$$

Here, Levenshtein$(s_1, s_2)$ calculates the Levenshtein distance between two strings, and $\min\{|s_1|, |s_2|\}$ returns the minimum length of the strings. This new value ranges between 0 and 1, with 0 indicating a perfect match between the strings and 1 implying that they were different at all positions.

The values in Table 2 show the Levenshtein distance between the true dataset header completion and the one made by the LLM. This demonstrates that the language model has completely memorised the first few rows of the Breast Cancer and Heart Disease datasets, shown by a normalised Levenshtein distance value of 0. This could be because the first few rows of these datasets commonly appear within the language model's training data. For the Diabetes and

*Table 2.* **GPT-3.5-turbo header completion test results.** Normalised Levenshtein distance between the correct header completion and the language model's completion for each of the datasets tested. A value of 0 indicates that the true completion and language model's completion were equal, and a value of 1 means that they were different at all positions.

| Heart Disease | Diabetes | Hypothyroid | Breast Cancer |
|---|---|---|---|
| 0.00 | 0.21 | 0.22 | 0.00 |

Hypothyroid datasets, we see some level of memorisation, with the language model being able to reproduce a large amount of the true header. As we might expect, when we perform the header test on the synthetic dataset generated using the relationship $y = 2x_1 - x_2 + x_3$, we get a normalised Levenshtein distance of $0.7$, which shows a large difference between the true header string and the completion from the language model. The UTI task has not been tested, as it is a private dataset.

*Table 3.* **GPT-3.5-turbo row completion test results.** Normalised Levenshtein distance between the correct row completion and the language model's completion for each of the datasets tested. A value of 0 indicates that the true completion and language model's completion were equal, and a value of 1 means that they were different at all positions.

| Heart Disease | Diabetes | Hypothyroid | Breast Cancer |
|---|---|---|---|
| 0.33 ± 0.05 | 0.38 ± 0.05 | 0.20 ± 0.05 | 0.43 ± 0.13 |

We supplement these results with those of Table 3, which shows the results of the row completion test. Here, we calculate the normalised Levenshtein distance of the true completions and the language model's completion of a random row from the dataset. In these tests, the language model is provided with 10 rows (in order from the original dataset) and is asked to complete the following row. The values in Table 3 represent the mean and standard deviation of the test using 25 random rows. Here, in contrast to the header test, we see that the language model has not memorised all rows from the dataset. This is because the language model's completion differs from the true row completion in all cases.

We likely see these results because when the datasets appear within the language model's training data, their column names and initial rows are usually given, but not the entire dataset.

### A.8.1. How Does this Affect our Results?

Within our experiments, we normalise the training and testing data and sample random observations from the dataset. Because of this, observations predicted during in-context learning will likely not have been memorised, as they are normalised based on a training data subset and are selected

from random rows of the dataset. Also, although the LLM can complete the first few rows of the datasets with some success, it is unable to complete random rows. Therefore, we hypothesise that any memorisation of the tested datasets has little impact on the in-context predictions.

Further, since for prior elicitation we only prompt the language model for a prior distribution of a predictive model given the feature names and a description of the dataset, the memorisation of rows from the data is unlikely to have had an impact on these results. In addition, we see that elicited priors provide improved predictions over the uninformative prior for the private and synthetic datasets (with no dataset memorisation), as well as the public datasets.

### A.9. Additional AutoElicit Results

#### A.9.1. Accuracy Distributions

In Figure 9, we present the accuracy distributions from the experiment in Figure 2, from Section 5.2.

For the synthetic task $y = 2x_1 - x_2 + x_3$, the priors elicited from the LLMs allow for a posterior predictive distribution with a markedly reduced mean squared error (MSE) over an uninformative prior whilst producing less variation in the result. Here, GPT-3.5-turbo initially produces the lowest MSE. However, due to the simple nature of the task, as more data becomes available, all priors achieve similar results.

When predicting UTI, all of the AutoElicit priors provide improved accuracy over the uninformative prior for all training sizes tested. This should be of particular interest, given that the UTI dataset is private and so will not have appeared in any form in the training data for the LLM.

For the Heart Disease dataset, we find that the prior elicited from the language model performed similarly to the uninformative prior. This is because the language model provided `mean = 0` and `std = 1` for a large number of priors elicited, suggesting that the LLM was unsure of its prior knowledge. The parameter distributions elicited can be seen in Figure 18, which contextualises these results. However, DeepSeek-R1-32B-Q4 and GPT-4-turbo provided prior distributions that were considerably more accurate.

For Hypothyroid prediction, DeepSeek-R1-32B-Q4 provides the best accuracy by a large margin. Of the OpenAI models, smaller training sizes favour GPT-4-turbo, whilst larger training sizes favour GPT-3.5-turbo. This is because the GPT-4-turbo-elicited prior was too strict to allow for updated posterior parameters, but was more informative than an uninformative prior for smaller training sizes.

For the Diabetes task, both OpenAI LLMs provide considerably improved MSE over the uninformative prior for training sizes up to 50 observed data points, whilst DeepSeek-R1-32B-Q4 provides worse MSE.

In the Breast Cancer task, the accuracy improvement from the elicited DeepSeek-R1-32B-Q4 and GPT-4-turbo priors is maintained across all training sizes, whilst GPT-3.5-turbo provides an improvement up to 40 observed data points. This can be seen in these distributions, which show a long tail in the accuracy distribution for the uninformative prior for most training sizes.

### A.9.2. PRIORS ELICITED FROM THE SYNTHETIC DATA

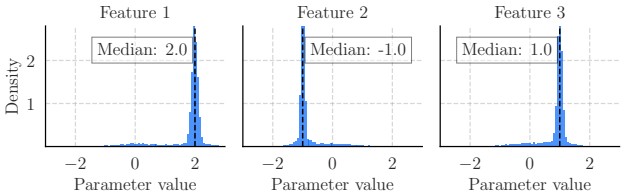

*Figure 10.* **Elicited priors for the task** $y = 2x_1 - x_2 + x_3$**.** We sample 10,000 parameter values from the mixture generated from the Gaussian priors elicited from the language model when given the task description. This figure shows the histograms of those values in each feature dimension.

Figure 10 shows the histogram of parameter values from the GPT-3.5-turbo-elicited prior for the task descriptions of $y = 2x_1 - x_2 + x_3$. Here, the language model correctly provides prior distributions that are centred around the correct parameter values and offer a small standard deviation.

### A.9.3. UTI ACCURACY WITH THE LABEL COLLECTION CONTEXT

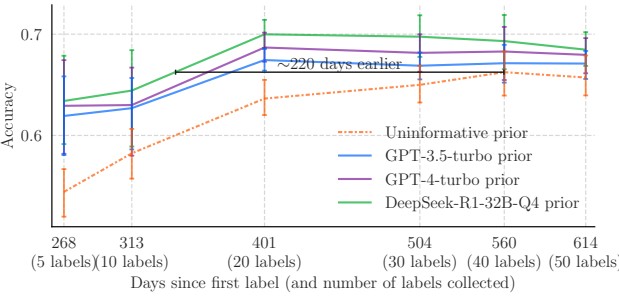

*Figure 11.* **Posterior accuracy of the elicited priors on the UTI dataset considering label collection time.** The average mean posterior accuracy and 95% confidence interval on the UTI test set after observing the given number of data points against the time it took for the data points to be collected in the study. The horizontal axis shows the time in days to achieve the given performance and the number of labels collected in that time.

Since we have access to the underlying study that collected the UTI dataset, we can understand in more detail how the accuracy of the predictive model varies in time, as well as with the number of data points collected.

In Figure 11, we plot the time it took to collect the given

number of labels tested in the experiments in Section 5.2 against the accuracy of a model using an uninformative prior and one elicited from a language model. This result illustrates the usefulness of prior elicitation, since we are able to achieve greater accuracy with fewer data points, and therefore less data collection time.

Since UTI label collection is costly and slow, our predictive model could have achieved greater accuracy significantly earlier in the study. This is paramount in a clinical environment, where an improved UTI detection model could have resulted in fewer unplanned hospital admissions.

It is important to note that the accuracy in Figure 11 is not achieved as a result of training on the earliest collected data points. This is because in our analysis, we perform cross-validation and use a subset of the data from a later time period (Section A.7.1). Changes in the in-home devices were made during the study, such as the addition of a device to collect sleep information, which we use within the dataset evaluated in the main results. However, if all data collection devices had been used from the beginning of the study, we could expect the accuracy in time to resemble the figure presented here.

### A.9.4. ANOTHER UNINFORMATIVE PRIOR

For additional context to the results provided in Figure 2, we present the posterior performance of a different uninformative prior. In this further case, we sample 100 values from a standard normal distribution ($\mu_k \sim \mathcal{N}(\mu|0, 1)$), and we use these as means of 100 Gaussian distributions ($\theta_k \sim \mathcal{N}(\theta|\mu_k, 1)$). With these distributions, we construct a mixture prior. As we keep the standard deviation at 1 for each component, the resulting uninformative prior has a greater range in the parameter space.

Figure 12 shows the results of this experiment compared to those achieved using AutoElicit. This new uninformative prior leads to no noticeable difference in the results for the synthetic task, Diabetes task, and Breast Cancer task. However, we see marginally better accuracy on the UTI task than the previous uninformative prior, and on the Heart Disease and Hypothyroid tasks, we see greater improvements. Despite this, DeepSeek-R1-32B-Q4 or GPT-3.5-turbo continue to provide more informative priors.

This suggests that the performance improvement we see from using AutoElicit over an uninformative prior is not due to its complexity.

### A.9.5. COMPARING WITH GOUK & GAO (2024)

The method presented by Gouk & Gao (2024) generates synthetic data for a predictive task using an LLM. Using this data, they provide an additional likelihood term when training linear classification models. Specifically, they use

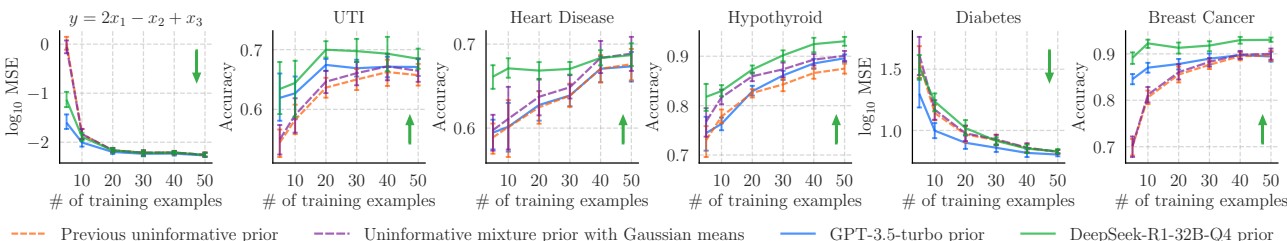

*Figure 12.* **Comparing AutoElicit to a mixture of uninformative priors.** This figure presents a linear model's posterior accuracy or mean squared error after observing the given number of training data points, over 10 repeats. Here, we compare the posterior performance of using AutoElicit with two LLMs to that of a mixture of uninformative priors, where the means of each component are sampled from a standard Gaussian distribution, as well as the uninformative prior presented in Figure 2.

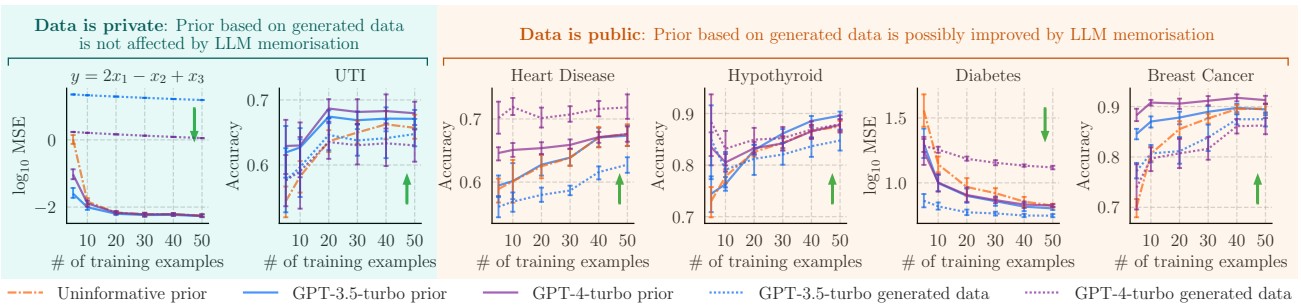

*Figure 13.* **AutoElicit and baseline from Gouk & Gao (2024).** This figure demonstrates a linear model's posterior accuracy or mean squared error after observing the given number of training data points over 10 repeats. The solid lines correspond to the results from AutoElicit, whilst the dashed lines correspond to the results from Gouk & Gao (2024). The left-most two datasets are unseen by an LLM, whilst the right-most four datasets are publicly available and likely have samples within the LLM's training data. In these cases, the LLM could be reproducing real data from the task, artificially improving the results for the method proposed by Gouk & Gao (2024).

an LLM to generate feature values based on a task description, which are labelled using zero-shot in-context learning. The samples generated are provided alongside real data during the training of a linear predictive model. However, this means that this method is susceptible to LLM data memorisation, putting it at risk of artificially providing improved posterior results on public datasets. Given this is a concern discussed in Bordt et al. (2024) and Appendix A.8, during our evaluation, we split the results by those achieved on publicly available tasks and those on privately available tasks. Additionally, to fully compare this method with AutoElicit, we extended their work to regression tasks. Figure 13 shows the results of this experiment.

In the public tasks, we see that the performance of the approach presented in Gouk & Gao (2024) varies considerably by LLM and task. It underperforms compared to the uninformative prior on five occasions. On the two occasions it outperforms AutoElicit (ours), it achieves surprisingly good accuracy or mean squared error (Heart Disease with GPT-4-turbo and Diabetes with GPT-3.5-turbo), suggesting that it might be reproducing parts of the public datasets rather than generating new samples. Given our analysis in Appendix A.8, showing the memorisation ability of LLMs, and the results of the method given in Gouk & Gao (2024) on pri-

vate data, we suspect the performance of this approach is artificially improved by reproducing the true dataset. Since for AutoElicit, we prompt the language model for a prior distribution over the parameters of a predictive model, using only the feature names and a description of the dataset, the memorisation of data points is unlikely to have had an impact on our method's results (Appendix A.8.1).

Moreover, in both of the private datasets (the two left-most plots), where there is no risk of LLM memorisation, AutoElicit results in improved posterior accuracy and mean squared error. On the synthetic task, our improvement is orders of magnitude better, and on the UTI task, our method significantly increases accuracy at all training sizes. In both cases, the approach proposed by Gouk & Gao (2024) underperforms compared to an uninformative prior.

### A.10. Cost of AutoElicit

The priors elicited from GPT-3.5-turbo and GPT-4-turbo incur a cost, as we use OpenAI's API. To elicit all of the priors used in the experiments in Section 5.2, we were charged $0.41 (on average $0.068 per dataset) for GPT-3.5-turbo and $7.56 (on average $1.26 per dataset) for GPT-4-turbo. As a rule, the more features a dataset contains or the longer

the task description, the more expensive priors are to elicit since the number of input and output tokens is greater. However, we did not make use of the batch API and instead made each API call sequentially. All experiments in this work could be implemented using the batch API since no new API call depends on a previous API result, which may significantly reduce costs[10].

Eliciting the DeepSeek-R1-32B-Q4-based prior distributions incurred no API cost, because the model is open-source and we were able to run it locally. However, to do this, we used an Nvidia A100, which will likely not be available in a low-resource setting. The API costs for the non-quantised model will apply in these cases[11].

### A.11. Task Descriptions with More Information

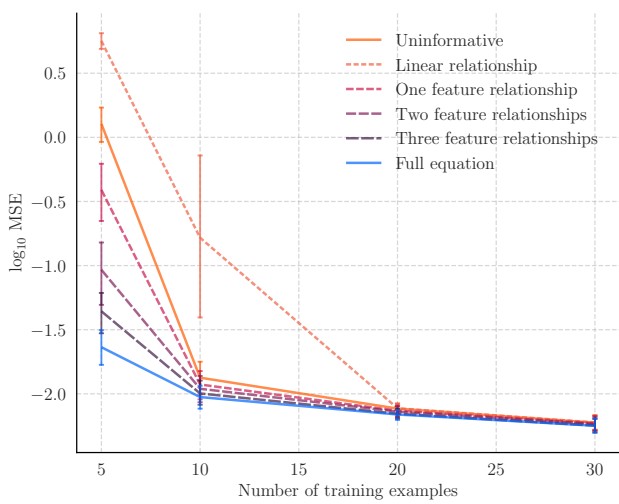

*Figure 14.* **Mean squared error using the GPT-3.5-turbo-elicited priors for the task:** $y = 2x_1 - x_2 + x_3$ **where we have used increasingly more detailed task descriptions.** Here, feature values are generated using $X \sim \mathcal{U}(-5, 5)$ and the targets are calculated using $y = 2x_1 - x_2 + x_3$. Within each task description, we provide the language model with increasingly more information about the task to test how the task description affects the priors elicited. Complete descriptions of the task descriptions given here are provided in Section A.11. The lines depict the average mean posterior mean squared error (MSE) over the 5 dataset splits, whilst the error bars represent the 95% confidence interval.

To understand how the detail of the task description affects the performance of the priors elicited from the language model, we studied the synthetic task $y = 2x_1 - x_2 + x_3$, and described the relationship between features and targets in increasing detail. We additionally test an uninformative prior to contextualise the results. For each of the levels of

detail shown in Figure 14, we included the following text in the task description that was provided to the language model:

- **Linear relationship:**
  "The target is linear in features"

- **One feature relationship:**
  "The target is a linear combination of the features and that when 'feature 0' increases by 1, the target increases by 2"

- **Two feature relationships:**
  "The target is a linear combination of the features and that when 'feature 0' increases by 1, the target increases by 2, and when 'feature 1' increases by 1, the target decreases by 1"

- **Three feature relationships:**
  "The target is a linear combination of the features and that when 'feature 0' increases by 1, the target increases by 2, and when 'feature 1' increases by 1, the target decreases by 1, and when 'feature 2' increases by 1, the target increases by 1"

- **Full equation:**
  "The 'target' = 2 * 'feature 0' - 1 * 'feature 1' + 1 * 'feature 2'"

For each prior elicited using these levels of detail, in Figure 14, the corresponding $\log_{10}$ MSE on the test set can be seen for a linear regressor after observing the given number of data points. This figure illustrates the effects of varied descriptions and shows that the improvements made to the MSE diminish as the task description contains more detail. The biggest improvement came from providing the language model with a single feature relationship – enabling an MSE that is two orders of magnitude better. This suggests that it is most helpful to provide more detail when the language model initially knows very little about a task.

Providing the language model with improved levels of detail allows it to produce prior distributions that enable our linear regression model to achieve a given error rate with fewer data points. As we might expect, the more detail provided in the task description, the better our prior distributions are.

However, it is unexpected that using an uninformative prior ($\theta \sim \mathcal{N}(\theta|0, 1)$) provides better posterior performance than specifying to the language model that the relationship between the target and features is linear (Figure 14). This is explained by inspecting the prior distribution elicited when given this minimal information, which shows that the language model responded with a similar mean and standard deviation for a Gaussian for each of the 100 variations of the prompts:

---

[10]https://openai.com/api/pricing/
[11]https://api-docs.deepseek.com/quick_start/pricing

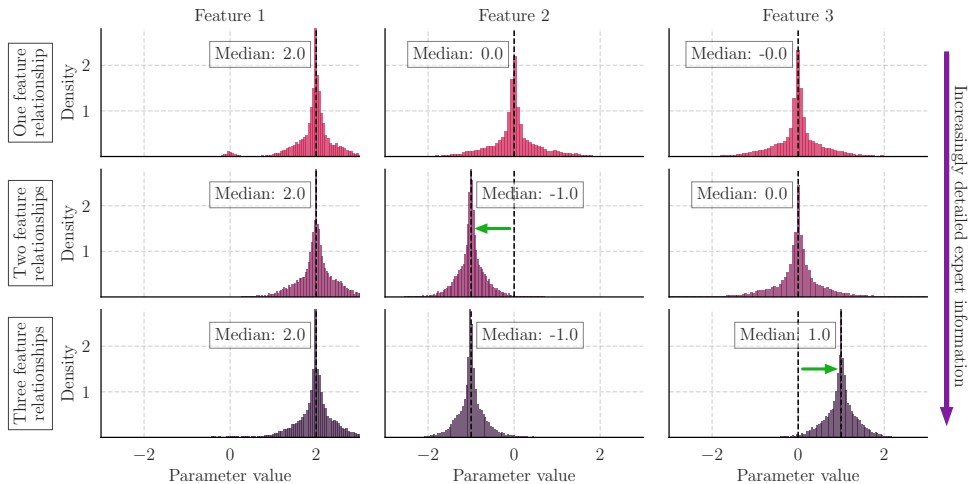

*Figure 15.* **GPT-3.5-turbo-elicited parameter distribution as increasingly more information is included.** These histograms show the distribution of parameter values for the features as more expert information is provided. In each row, we provide information in the task description of an additional feature, with the green arrow indicating the effect this has on the parameter distribution. In particular, in the first row we provide information about the relationship between the target and the first feature, in the second row we provide information about the second feature, and in the third row we provide information about the third feature.

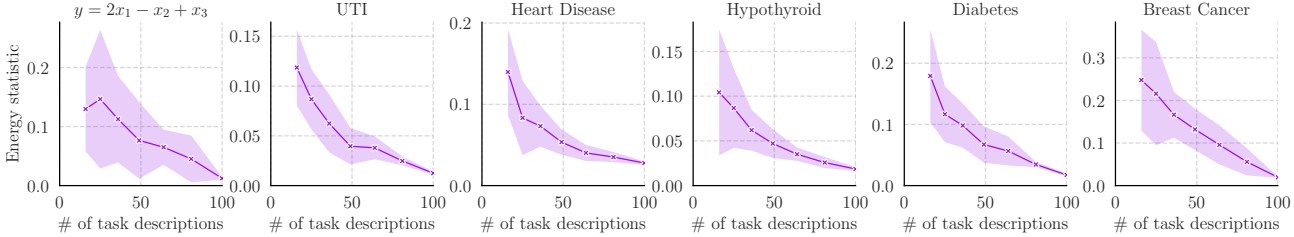

*Figure 16.* **Measuring how similar the GPT-3.5-turbo-elicited prior distributions are with fewer task descriptions.** We reduce the number of task descriptions and calculate the energy statistic between 10,000 samples from the prior elicited from the language model using the reduced set of descriptions and the full 100 task descriptions. We repeat each experiment 10 times and show the mean and standard deviation.

- Feature 0: mean: 0.5 ± 0.11, std: 0.12 ± 0.05

- Feature 1: mean: -0.32 ± 0.08, std: 0.08 ± 0.06

- Feature 2: mean: 0.56 ± 0.26, std: 0.14 ± 0.09

This demonstrates the sometimes unexpected behaviour of language models. Here, the LLM produced Gaussians for the prior distribution with small standard deviations (reflecting a high confidence in the distribution) and means that the language model knew the underlying equation defining the relationship between the target and features. We might have expected that after only being told the relationship was linear, the language model would produce Gaussians with 0 mean and standard deviation 1, reflecting the uncertainty in the exact relationship between each feature and the target.

Interestingly, although "Three feature relationships" and "Full equation" contain the same information written differently, the language model provided better prior distributions

when given the full equation. Fully specifying the equation is not realistic in a real-world setting where we are unsure about the exact relationship between features and targets, but this provides an example of how we can make the task description as information-dense as possible.

An expert interacting with the language model might be able to provide further information about a task that improves the elicited priors. In Figure 15, we show the parameter distributions elicited from the language model for the different descriptions we provide. As we provide additional information on each feature, the language model responds by providing a prior distribution that reflects its updated knowledge of the task. Since the language model knows nothing about this synthetic task beforehand, this experiment demonstrates the effectiveness of using the AutoElicit framework to convert expert information as natural language into prior probability distributions. This experiment provides a controlled setting that supplements the results in Section 5.3.

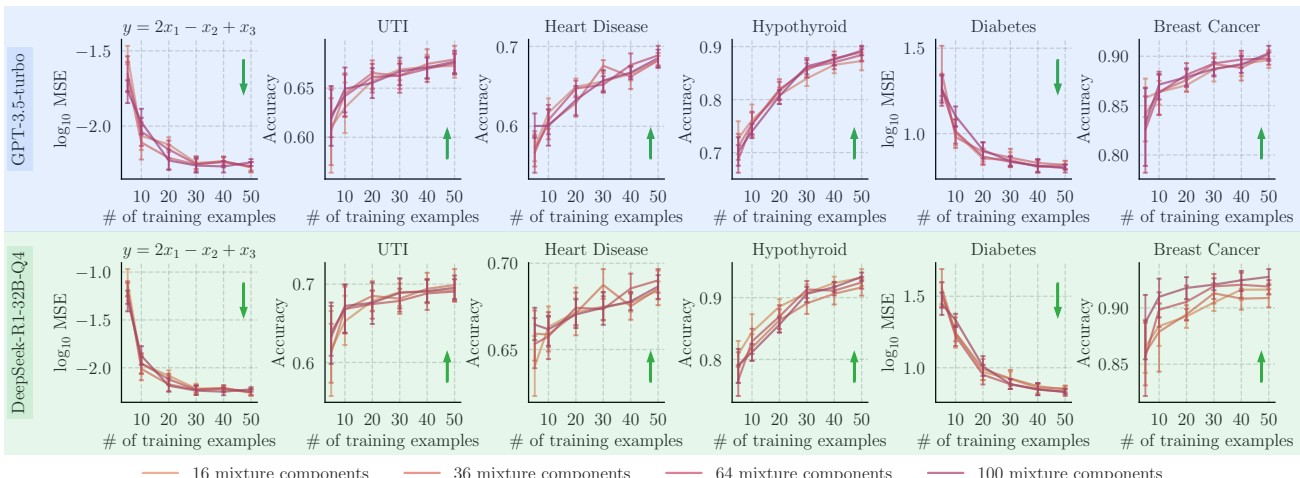

*Figure 17.* **AutoElicit with different numbers of mixture components in the prior.** This figure demonstrates a linear model's posterior accuracy or mean squared error after observing the given number of training data points, over 10 repeats. The rows of the figure correspond to the different language models tested, whilst the colour of the lines indicates the number of mixture components used in the prior. Using 16 mixture components corresponds to using a product of 4 system roles and 4 user roles when eliciting priors.

## A.12. How Many Ways Should We Describe a Task?

In this section, we examine how many task descriptions are required to be able to get a similar prior distribution to that elicited within our experiments, using 100 task descriptions.

Figure 16 shows the energy statistic (Appendix A.5) of samples from the prior distribution elicited using fewer task descriptions compared to samples from the prior distribution elicited using 100 task descriptions (from the same language model). In particular, we compare a subset of task descriptions with the full 100 task descriptions.

We see that for most tasks, a large number of task descriptions is required, since reducing the number of descriptions from 100 to 49 produced prior distributions that are significantly different for all datasets tested. Using 81 task descriptions produces prior distributions that are more similar but still show differences for some datasets (for example, $y = 2x_1 - x_2 + x_3$ and Breast Cancer).

It is interesting that Breast Cancer benefited the most from increasing the number of task descriptions, since this dataset has the largest number of features. It suggests that more task descriptions are required for datasets with high-dimensional data. This makes intuitive sense since we might expect a larger mixture of Gaussian components to better describe the prior over for a higher-dimensional dataset.

Further work could allow the number of task descriptions to be a function of the number of features in the dataset.

To supplement this, we calculate the posterior performance with varied numbers of mixture components in our prior, shown in Figure 17. We see little variation (except for Breast

Cancer with DeepSeek-R1-32B-Q4) in the accuracy or mean squared error as we increase the number of components in the prior, suggesting our framework is robust to the number of task descriptions. We hypothesise that a greater number of mixture components makes the framework more resilient to LLMs that frequently hallucinate, especially on tasks with large numbers of features (such as Breast Cancer).

Figure 16 showed that the density of the priors changed with the number of mixture components, however, this has a small effect on the posterior performance. This is likely because the sample space that provides good posterior performance is covered by the majority of elicited components, whilst the large changes in Figure 16 are due to more unique components.

## A.13. Are AutoElicit Priors Interpretable?

As the density of the distributions in Figure 18 are used for the prior directly, they allow for interpretable predictions. Considering the AutoElicit priors, for linear models, each feature's prior corresponds to its direction and importance when predicting the target. On the synthetic task (row 1), the prior distributions concentrate on the parameters that we expect. For the UTI task (row 2), the LLM provided prior distributions that suggest Feature 2 was negatively predictive of a positive UTI, whilst Feature 10 is a strong positive predictor. For Breast Cancer (row 6), the LLM produced two modes in its Feature 2 prior. Here, the LLM was either unsure of its belief, and so the elicited prior changed between task descriptions, or the LLM was unsure of the feature's direction of correlation but guessed that the feature was important.

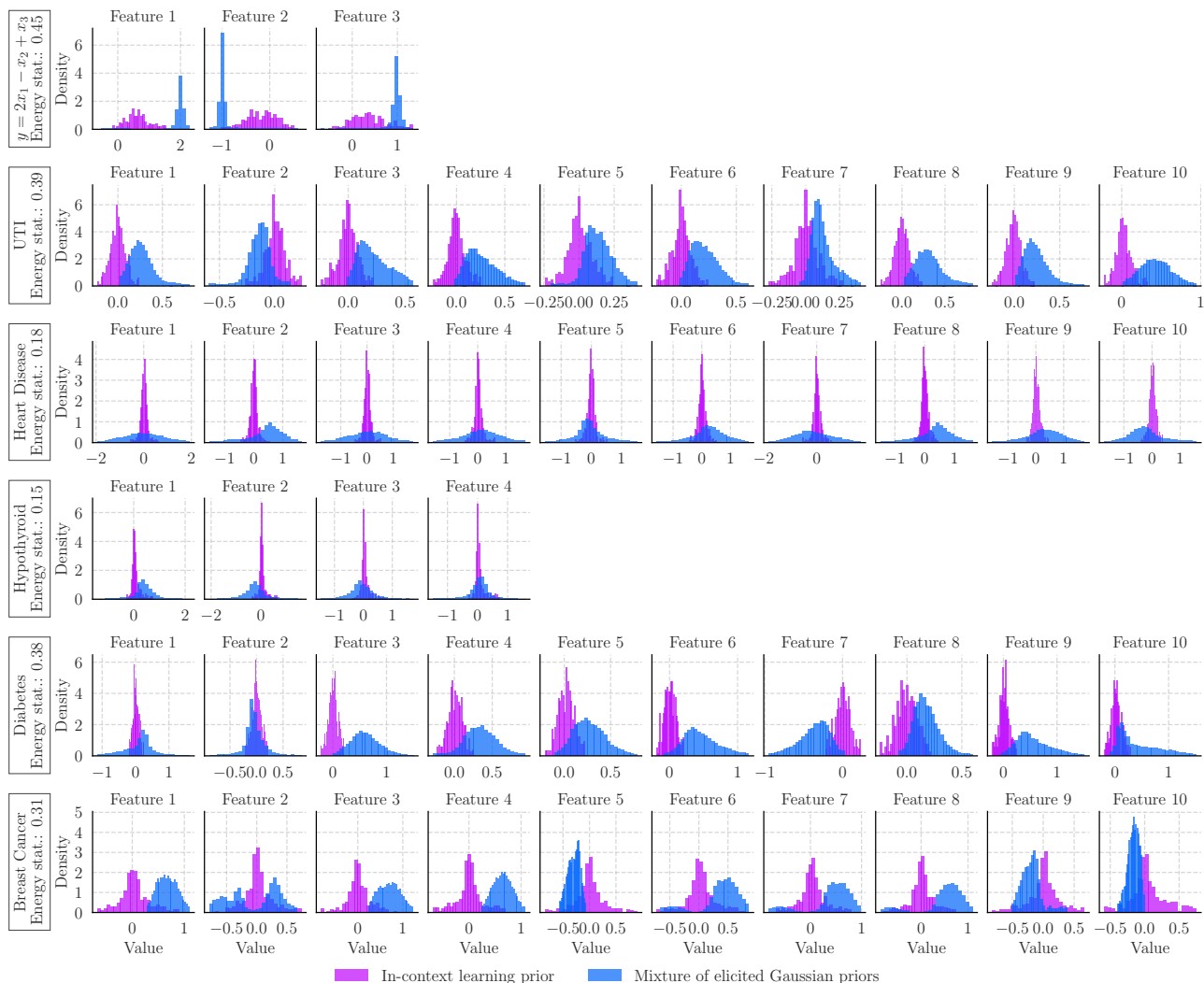

*Figure 18.* **Distribution of the prior parameters under elicitation and those approximated from the internal model for the different predictive tasks for GPT-3.5-turbo.** Here, we compare the prior distributions of the different predictive tasks, as defined by the internal model used in-context, and that which we elicit from the language model. Here, we draw the histogram separately in each dimension, but calculate the energy distance across all features combined.

A chain-of-thought LLM such as DeepSeek-R1-32B-Q4 could help us better understand the elicitation process.

## A.14. Is the Language Model Being Consistent During In-context Learning?

In Section 5.5, we study how consistent GPT-3.5-turbo is when we elicit a prior distribution, compared to the prior it uses in its internal predictive model when performing in-context learning.

Figure 18 visualises the different prior parameter distributions for all of the tested datasets in each feature dimension. Notable are the large differences between the elicited prior distributions and the ones used when performing in-context

learning, which supports the findings in Section 5.5. In the in-context case, the prior distributions often have modes close to 0, suggesting that the internal model is not sure about the relationship between features and targets. In contrast, the elicited priors often represent a stronger belief in the correlation between features and targets, with most histograms achieving modes greater than or less than 0.

Interestingly, on the Breast Cancer dataset, prior elicitation resulted in two distinct modes in the prior distribution for Feature 2 ("mean texture"), which could represent uncertainty in the direction of correlation.

Furthermore, the prior distributions on the parameters for the Heart Disease and Hypothyroid datasets have more variation

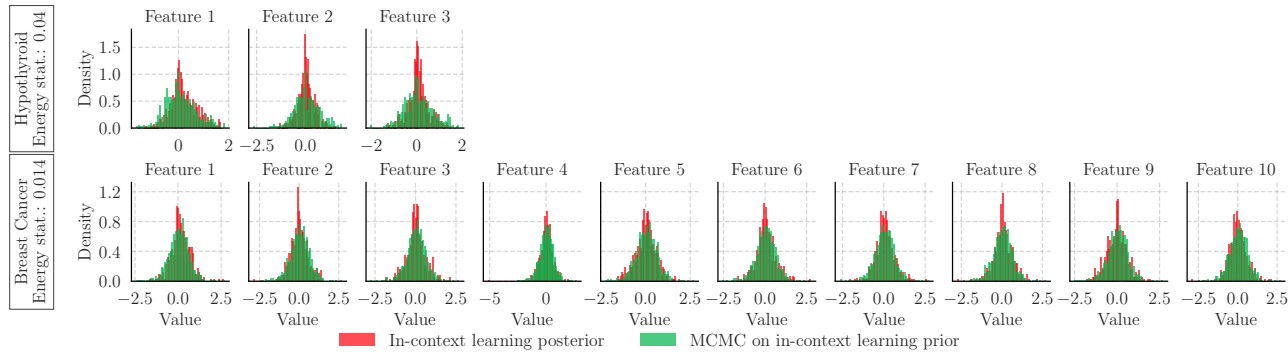

*Figure 19.* **The different posterior samples for GPT-3.5-turbo.** Here, we compare the distribution of the posterior extracted from the language model's in-context predictions against the distribution that we sample using Monte Carlo methods on the extracted prior distribution. Both posterior distributions are based on the same 25 data point sample from the corresponding dataset. We draw the histogram separately in each dimension, but calculate the energy distance across all features combined.

in their values in comparison to that extracted from the language model's internal model. This tells us that the elicited prior parameters represent more uncertainty in their value and are therefore more flexible when calculating the posterior on observed data.

## A.15. Empirically, Is the Language Model Approximating Bayesian Inference?

In this section, we present the differences between the posterior distribution extracted through maximum likelihood sampling of GPT-3.5-turbo's in-context internal model (Section 5.4) and one that is sampled through MC methods on the extracted in-context prior distribution.

*Table 4.* **Difference between approximated posterior and MC on the approximated prior for GPT-3.5-turbo.** The energy statistic between the language model's internal posterior extracted through MLE sampling and the posterior samples from applying Monte Carlo techniques to the language model's internal prior. Values are $\in [0, 1]$ with values closer to zero, indicating that the two distributions are more similar. Here, we show the mean and standard deviation of the energy statistic for the 5 different training sets tested as observed data.

| Heart Disease | Hypothyroid | Diabetes | Breast Cancer |
|---|---|---|---|
| $0.04 \pm 0.01$ | $0.06 \pm 0.01$ | $0.02 \pm 0.00$ | $0.01 \pm 0.00$ |

Table 4 shows the mean and standard deviation of the energy distance between these distributions, for all of the datasets tested, including those that have a high approximation error when we estimate the in-context posterior. In addition, Figure 19 shows the distribution of these posterior parameters for visual inspection.

For the Breast Cancer dataset, these two distributions match closely and show a strong agreement in their parameter values, suggesting that for this dataset, in-context learning

could be performing Bayesian inference. However, this should be considered alongside the results in Section 5.4, where we found that our approximations of the posterior distribution for the Breast Cancer task had a large error.

On the other hand, for the Hypothyroid predictive task, the parameter distributions differed significantly, with the in-context prior having a less varied distribution. This is reflected in the energy distance, which is the largest of all of the datasets tested.

These comparisons must be made with the approximation error in mind, presented in Section 5.4, which suggests that the posterior in-context distributions for the Breast Cancer and Hypothyroid tasks are not reliable.

## A.16. AutoElicit with Other Language Models

In this section, we will compare the results presented in the main paper with other commonly used language models. We start by exploring the performance of some other OpenAI models, before considering some of the open-source Llama and Qwen models.

### A.16.1. OTHER OPENAI MODELS

Here, we compare the results of priors elicited from GPT-3.5-turbo, DeepSeek-R1-32B-Q4 (open-source), and GPT-4-turbo (which incurs a significant cost) to GPT-4o-mini (which provides a less expensive option).

To elicit all priors presented here, GPT-4o-mini cost $0.32 (on average $0.053 per dataset), whilst GPT-4-turbo cost $7.56 (on average $1.26 per dataset) and GPT-3.5-turbo cost $0.41 (on average $0.068 per dataset).

Figure 20 shows that GPT-4o-mini often produces prior distributions that are equal to or worse than GPT-3.5-turbo in their posterior performance on all tasks except one. For

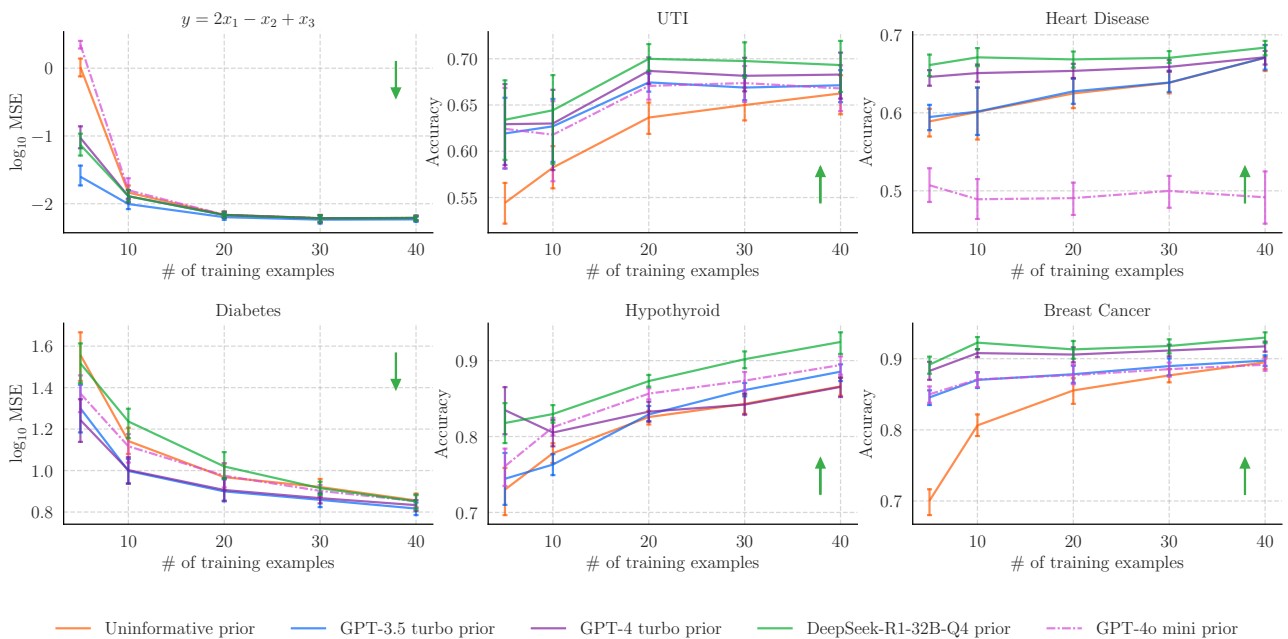

*Figure 20.* **Test accuracy of the posterior distribution compared to GPT-4o mini.** Here we show the posterior results when using priors elicited from two other OpenAI language models and an uninformative prior $\mathcal{N}(\theta|0, 1)$. These results are calculated on a test set of the datasets after each model is trained on the given number of data points. The values are the result of 5000 sampled parameters for each of 5 chains, and 10 random splits of the training and testing data. For the regression tasks, we report the mean squared error on a log scale. This figure shows the line plot of the average mean posterior accuracy or mean squared error over the 10 splits, with the error bars representing the 95% confidence interval. The green arrows point in the direction of improvement in the metric.

the synthetic, Diabetes, and Heart Disease prediction tasks, GPT-4o-mini provides priors that achieve similar or worse performance to the *uninformative* prior. However, for the UTI and Breast Cancer predictive tasks, they achieve similar posterior accuracy to GPT-3.5-turbo at a lower price point. Also, on the Hypothyroid dataset, GPT-4o-mini provides prior distributions that achieve greater accuracy than those from GPT-3.5-turbo and GPT-4-turbo for all training sizes, whilst requiring significantly less cost. Further, in all tasks except Diabetes progression prediction, DeepSeek-R1-32B-Q4 provides better accuracy than GPT-4o-mini.

### A.16.2. OPEN-SOURCE LANGUAGE MODELS

Here, we explore the posterior performance of prior distributions elicited from three Llama (Touvron et al., 2023; MetaAI, 2024; Dubey et al., 2024) models, one Qwen model (Yang et al., 2024; Team, 2024), and one DeepSeek model (DeepSeek-AI et al., 2025) to those elicited from the OpenAI models presented in the main paper:

- DeepSeek-R1-32B-Q4[12], a 32 billion parameter model from DeepSeek based on Qwen 32B and distilled from DeepSeek-R1.

---
[12]https://ollama.com/library/deepseek-r1:
32b

- Llama-3.2-3B-Instruct[13], a 3 billion parameter model from Meta's latest collection of v3.2 Llama models.

- Llama-3.1-8B-Instruct[14], an 8 billion parameter model from Meta's v3.1 Llama collection.

- Llama-3.1-70B-Instruct-BF16[15], a bfloat16 quantisation of Llama-3.1-70B-Instruct, a 70 billion parameter model from Meta's v3.1 Llama collection.

- Qwen-2.5-14B-Instruct[16], a 14 billion parameter model from Alibaba Cloud's v2.5 Qwen collection.

In contrast to the OpenAI models in Appendix A.16.1, these models are publicly available and open-source. This means that they can be used without the need for connecting to an online API, and when locally hosted, all data and inference can be performed on-device, enabling the use of our framework within a closed and secure environment.

---
[13]https://huggingface.co/meta-llama/
llama-3.2-3b-instruct
[14]https://huggingface.co/meta-llama/
llama-3.1-8b-instruct
[15]https://huggingface.co/meta-llama/
llama-3.1-70b-instruct
[16]https://huggingface.co/qwen/qwen2.
5-14b-instruct

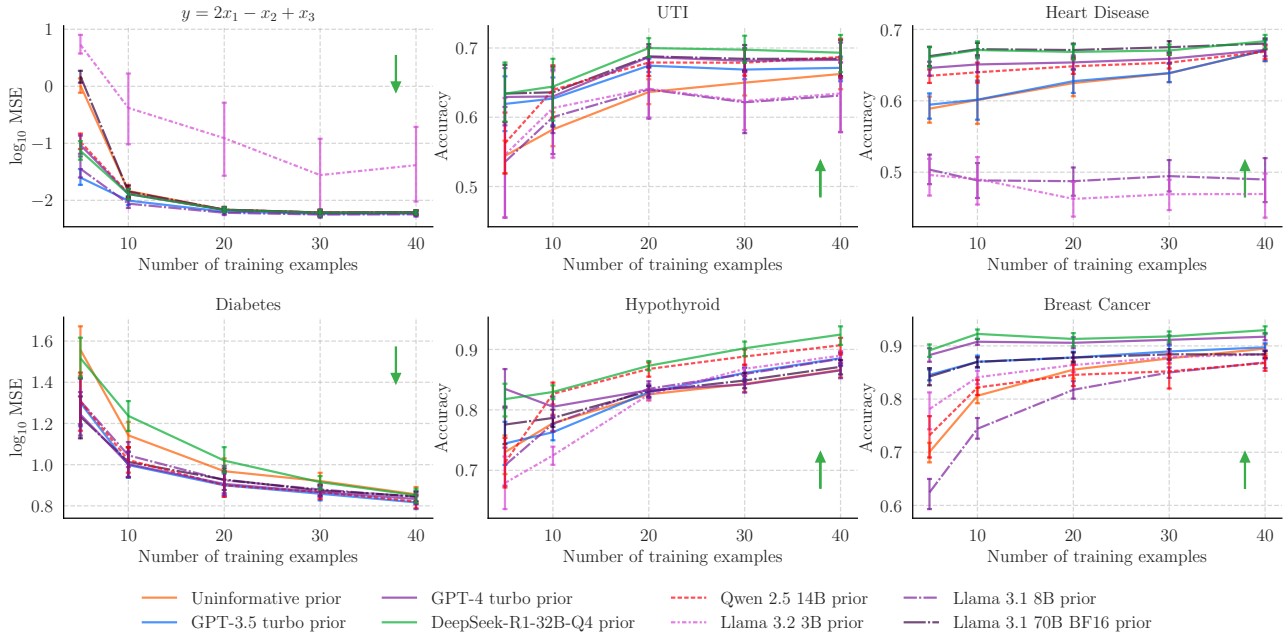

*Figure 21.* **Test accuracy of the posterior distribution for open-source language models.** Here, we show the results when using priors elicited from two of the Llama language models and an uninformative prior $\mathcal{N}(\theta|0, 1)$ on a linear model. These results are calculated on a test set of the datasets after each model is trained on the given number of data points. The values are the result of 5000 sampled parameters for each of 5 chains, and 10 random splits of the training and testing data. For the regression tasks, we report the mean squared error on a log scale. This figure shows the line plot of the average mean posterior accuracy or mean squared error over the 10 splits, with the error bars representing the 95% confidence interval. The green arrows point in the direction of improvement in the metric.

The experiments presented in this section use the same task descriptions as given to GPT-3.5-turbo, GPT-4-turbo and DeepSeek-R1-32B-Q4. When sampling the posterior with the elicited priors, we use the same methods and options as given in Appendix A.7.3.

Figure 21 illustrates the inconsistent posterior accuracy of the priors elicited from the Llama family of models. The 3B and 8B variants are significantly smaller in capacity to those in the main paper, justifying their inconsistent elicited priors. However, in some of the datasets, they still produce good posterior performance.

Specifically for Breast Cancer, the 3B parameter Llama model provides an intermediate accuracy between an uninformative prior and those elicited from GPT-3.5-turbo, and for the synthetic and Hypothyroid tasks, the 8B parameter Llama provides priors of equal quality to GPT-3.5-turbo. However, for the UTI and Heart Disease datasets, both the 3B and 8B Llama models provide prior distributions that are of worse quality than even the uninformative prior. It is possibly not surprising that Llama-3.2-3B-Instruct and Llama-3.1-8B-Instruct provided priors that were equal or worse than Llama-3.1-70B-Instruct-BF16 in all cases except for the synthetic task, since these models are significantly smaller in size.

Llama-3.1-70B-Instruct-BF16 is more competitive, sometimes providing priors that allow for better posterior distributions than GPT-3.5-turbo, but on other occasions, providing priors with posterior performance similar to the uninformative prior. On the UTI and Heart Disease datasets, the prior distributions elicited from Llama-3.1-70B-Instruct-BF16 provide better posterior accuracy than those from GPT-3.5-turbo and comparable values to GPT-4 turbo or DeepSeek-R1-32B-Q4. Surprisingly, on Heart Disease prediction in particular, priors from the 70B Llama model provided a posterior accuracy greater than GPT-4-turbo and equal to DeepSeek-R1-32B-Q4. Additionally, when predicting Breast Cancer, priors from Llama-3.1-70B-Instruct-BF16 achieved the same posterior accuracy as GPT-3.5-turbo for all of the training sizes tested. These three datasets suggest that Llama-3.1-70B-Instruct-BF16 can be a useful model for eliciting prior distributions for predictive tasks.

Qwen-2.5-14B-Instruct produces priors with similar inconsistent posterior performance. For Heart Disease and Hypothyroid prediction, it achieves a competitive posterior accuracy, which is surprising given its size. For the UTI task, after seeing 10 demonstrations, it performs comparably to Llama-3.1-70B-Instruct-BF16 and GPT-4-turbo, and better than GPT-3.5-turbo. However, on the Breast Cancer and synthetic tasks, it performs worse than GPT-3.5-turbo, but is

likely smaller in capacity. Overall, Qwen-2.5-14B-Instruct provides inconsistent performance across tasks, but has the benefit of performing similarly to Llama-3.1-70B-Instruct-BF16 on many tasks with a significantly smaller compute requirement.

The fact that DeepSeek-R1-32B-Q4 is an intermediate size between Qwen-2.5-14B-Instruct and Llama-3.1-70B-Instruct-BF16, and provides considerably better performance with a higher quantisation is a testament to the power of recent advances in reasoning (Huang & Chang, 2023) and test-time scaling (Jones, 2021).

Therefore, it is important, given an elicited prior distribution, to calculate the prior likelihood with some already collected data to understand which prior is most appropriate for constructing a predictive model.

### A.17. Choosing Between Priors

As seen in Appendices A.16.1 and A.16.2, the prior distributions elicited from language models do not always provide the best posterior performance. We consistently find that DeepSeek-R1-32B-Q4, GPT-3.5-turbo and GPT-4-turbo provide better prior distributions than an uninformative prior; however, this may not be the case for all datasets or language models. Therefore, given prior distributions elicited from different language models, we would like to choose the best one for our predictive task.

For this, given some collected data, we can use the Bayes factor (Section 4.3) as a model selection technique to decide between the prior distributions elicited from different language models. Alternatively, we could use other model selection techniques, such as the Bayesian Information Criterion.

Further, by using a temperature parameter, it is also possible to control the strength of the elicited prior, which can indicate our trust in the distribution.

