# OpenReview forum: "AutoElicit: Using Large Language Models for Expert Prior Elicitation in Predictive Modelling"
_ICML.cc/2025/Conference — ICML 2025 poster_

### Official Review · Reviewer_F25r · 2025-03-03

**Overall Recommendation:** 4

**Summary:**

This paper explores the potential of LLMs to elicit prior distributions, aiming to reduce sample complexity in Bayesian inference, particularly in data-scarce healthcare domains. The authors compare two LLM-based prior elicitation methods: their proposed AutoElicit and In-Context Prior. AutoElicit directly prompts the LLM to provide estimated means and variances for Gaussian priors in text. In contrast, In-Context Prior elicits posterior predictive distributions from the LLM, from which implicit priors are inferred using maximum likelihood. The authors demonstrate that both methods can be beneficial across various tasks, with AutoElicit generally yielding superior priors. Notably, the paper shows clear improvements in inference when leveraging LLM-elicited priors compared to uninformative priors. Furthermore, the methodology facilitates the incorporation of expert knowledge expressed in natural language during the prior elicitation process.

**Claims And Evidence:**

This paper presents commendable work, offering straightforward and effective methods. The manuscript is well-written and easy to follow. The application to healthcare domains is highly relevant, and the extensive experiments demonstrate the utility of LLM-elicited priors.

Considering the significance of the healthcare application, a deeper mechanistic understanding of the LLM's reasoning in generating these effective priors would be valuable. While space limitations might preclude a full mechanistic analysis, a discussion exploring potential explanations for the LLM's success in this context would strengthen the paper.

**Essential References Not Discussed:**

I don't see any other essential reference that should be included.

**Ethical Review Concerns:**

The healthcare datasets presented within this paper raise potential ethical concerns and likely require independent ethical review.

++++

Details of ethic review below:

Regarding research ethics: the empirical evaluation uses one synthetic data set, four public data sets, and one (sensitive) data set collected by the authors (which they label "private"). From Section 5.1, para.4 of the paper:

"We also evaluate AutoElicit on a private dataset not in the LLM’s training, collected as part of our study on Dementia care. This contains 10 features of daily in-home activity and physiology data, and clinically validated labels of positive or negative urinary tract infection (UTI), based on the analysis of a urine sample by a healthcare professional. This dataset contains 78 people and 371 positive or negative days of UTI."

However, the authors provide no statement/evidence that appropriate IRB approvals were obtained or protocols followed. I would like the authors to briefly state whether this is the case in their response. This is in accordance with the ICML guidelines: "While we acknowledge that standards for IRB vary across borders and institutions, research involving human subjects should provide evidence that it adhered to the authors’ home institution’s procedures for obtaining IRB approval or was eligible for an exemption."

**Ethical Review Flag:**

Flag this paper for an ethics review.

**Ethics Expertise Needed:**

["Privacy and Security", "Responsible Research Practice (e.g., IRB, documentation, research ethics, participant consent)"]

**Experimental Designs Or Analyses:**

I have reviewed all their experiments.

**Methods And Evaluation Criteria:**

The authors tested the LLM-elicited priors (along with uninformative priors) on synthetic data and healthcare datasets. I find their evaluation and methods adequate.

**Other Comments Or Suggestions:**

None.

**Other Strengths And Weaknesses:**

None.

**Questions For Authors:**

Have you tried eliciting priors using the iterated in-context learning proposed in Zhu & Griffiths (2024)? How would the priors elicited using this approach compare to those elicited using AutoElicit and In-Context Priors?

**Relation To Broader Scientific Literature:**

This work is also related to other work that tries to link LLM (and in-context learning) and Bayesian inference.

**Theoretical Claims:**

No theoretical claim was made in this paper.

---

> ### Author Rebuttal · Authors · 2025-03-31
>
> We would like to thank the reviewer for their time and comments to improve our paper. We are pleased that they felt that this was commendable work that is well-written, easy to follow, and highly relevant for healthcare. We are glad that the methods were considered straight-forward, but effective, and that reviewer felt that the extensive experiments demonstrated the utility of our approach.
>
> In our rebuttals, alongside our point-by-point responses, we present **three new experiments**.
>
> ### **Suggested improvements**
>
> **Q1: A potential explanation for the LLM's success**
>
> We believe our method’s success is due to the LLM’s ability to translate information across domains. LLMs can effectively answer questions in medical domains and probability theory. They offer the ability to translate expert information in the form of natural language into probability distributions that represent prior beliefs on model parameters. We hypothesise that resampling the LLM multiple times mitigates the effects of hallucination for LLMs that are more likely to fabricate information. By combining many Gaussian components, we can approximate an LLM’s non-Gaussian prior. For our method to be successful, the LLM needs to have an understanding of the problem domain, Gaussian distributions, and linear models. We believe that this is why we see improved performance from chain-of-thought models, which can reason about the impact of each feature on an outcome. We will include this discussion in the final version of the work.
>
> **Q2: Further comparisons to a baseline**
>
> Zhu & Griffiths (2024) is not directly comparable to our work, as they do not provide a framework for eliciting the parameters of a predictive model. The most comparable work, Gouk & Gao (2024) (cited), generates synthetic data using an LLM to include in the training of linear *classification* models. To compare with our method, we extend their ideas to *regression* tasks and present new experimental results.
>
> These are presented in our response to **Reviewer 1 (xAxP) in section Q2**, along with a detailed description of the baseline. In particular, Gouk & Gao’s method is susceptible to LLM data memorisation, putting it at risk of artificially providing improved posterior results on public datasets. To better compare the methods, we have split the results by the private and public tasks. We find that our approach yields improved posterior accuracy and mean squared error over Gouk & Gao’s method.
>
> **Q3: Add ethical approval statement**
>
> We have already obtained the appropriate ethics for our study but wanted to meet the double-blind requirement by not providing exact information on the ethics approval, including the identification number for our study. We include this information in the final version, alongside a description of the recruitment process. In the ethics statement, we have redacted the appropriate information to prevent breaking the double-blind requirements:
>
> > This study was performed in collaboration with INSTITUTION. Participants were recruited from the following: (1) health and social care partners within the INSTITUTION, (2) urgent and acute care services within the INSTITUTION, (3) social services who oversee sheltered and extra care sheltered housing schemes, (4) INSTITUTION and (5) specialist memory services at INSTITUTION. All participants provided written informed consent. Capacity to consent was assessed according to Good Clinical Practice, as detailed in the FRAMEWORK and the ACT. Participants were provided with a Participant Information Sheet (PIS) that includes information on how the study used their personal data collected in accordance with the ACT requirements. If the participant was deemed to lack capacity, a personal or professional consultee was sought to provide written consent to the study. Additionally, the capacity of both the participant and study partner is assessed at each research visit. Research staff conducting the assessment have completed the COURSE training and COURSE training. If a participant is deemed to lack capacity but is willing to take part in the research, a personal consultee is sought in the first instance to sign a declaration of consent. If no personal consultee can be found, a professional consultee, such as a key worker, is sought. This process is included in the study protocol, and ethical panel approval is obtained.
>
> We feel the use of privately collected data from a real-world healthcare problem provides substantial evidence for the effectiveness of our framework, and we are indebted to our study participants.
>
> **Thank you for taking the time to review our work. We will implement the suggested improvements to the final manuscript. If we have answered your questions, then we would appreciate you considering raising your score. If anything is still unclear, we are happy to discuss further.**

---

> > ### Comment · Reviewer_F25r · 2025-04-01
> >
> > I thank the authors for their thoughtful responses. I will keep my score unchanged, as it already reflects strong support for accepting this paper.

---

> > > ### Author Response · Authors · 2025-04-02
> > >
> > > We thank the reviewer for their response and time. Their comments have helped to improve our work.

---

### Official Review · Reviewer_fJAR · 2025-03-13

**Overall Recommendation:** 2

**Summary:**

This work proposes a framework to elicit prior distributions over parameters from LLMs, and use these in connection with linear models, with the goal to achieve transparent and interpretable, yet performant models.

**Claims And Evidence:**

The paper is well written and relatively easy to follow.

**Essential References Not Discussed:**

In Sec. 5.6 the authors study the question whether in-context learning in LLMs is Bayesian. The authors also mention this as one of the three contributions in Sec. 1. However, this question has been studied already. Some of these papers you have cited in App. 2.2. (though not in the main text, why?). It may be worth adding “Are Large Language Models Bayesian? A Martingale Perspective on In-Context Learning” (ICML 2024).

**Experimental Designs Or Analyses:**

The experiments are strong, I particularly like the idea of including expert information in addition. The figures are extensive, informative and clear.

The basic setup and notation are simple, yet somewhat hard to follow. An example explaining important concepts such as task description $I_k$ (e.g. App. 7.3) clearer.

**Methods And Evaluation Criteria:**

The application settings range from synthetic to real-world clinical (private) dataset, which I very much appreciate. This renders the experimental setup interesting and relevant.

**Other Comments Or Suggestions:**

N/A

**Other Strengths And Weaknesses:**

- The appendix is detailed, and very helpful in augmenting the main text.

**Questions For Authors:**

- Have you tried other uninformative priors than N(0,I), even a higher variance for instance? Given you use a K=100 MoG, a natural idea might be to try a MoG prior possibly with randomised parameters. -- What I would be interested in disentangling is the question whether the complexity of the prior (MoG vs. single Gaussian in the uninformative case) or the “correctness” of the LLM prior is crucial for performance.

- Can you state how you compute predictive performance? What’s the predictive distribution you use / can you point me to the equation?

**Relation To Broader Scientific Literature:**

N/A

**Theoretical Claims:**

Major weakness:

My main point of criticism is on the loss of *transparency/interpretability* of the proposed approach: *Context*: The auto-elicited prior is constructed as a Mixture of Gaussians (see Eq. 2). In the experiments, the authors choose K=100 components, and in App. 12 the authors show that this very complex prior is necessary to achieve strong predictive performance.  *Criticism* Is it desirable to construct such complex prior distributions with LLMs and then apply them to linear models with the goal of maintaining interpretability? The goal of using a linear model is two-fold: 1) the functional relationship is simple and ‘interpretable’ (this is maintained with the proposed approach) 2) the weights are fitted by a simple least-squares error procedure (or a similarly simplistic and transparent procedure here in the Bayesian case). If the prior now is extremely complicated, both because it contains knowledge of the LLM and it is a 100-component MoG, we lose all transparency of the predictive weights and point 2) is lost/’violated’. Because of this, I cannot see a clinician trusting the model as an “interpretable”/simple model anymore. We have---through the rules of Bayesian Inference---made our linear model a linear model augmented with an LLM. If we wanted to improve the performance of the a (linear) model, we could use an LLM directly, or use another model. -- Am I missing something in terms of interpretability? I think it’s important to discuss this point in the rebuttal; at this point, it drastically diminishes the significance of the work for me. -- Also, I would like to note that the general procedure of “eliciting priors from LLMs” remains interesting (though has been done in related work), it is only the use for predictive purposes in the context of trying to maintain the interpretability of linear models that I find challenging.

I would like to discuss this point during the author rebuttal and will revisit my decision based on this discussion.

---

> ### Author Rebuttal · Authors · 2025-03-31
>
> We would like to thank the reviewers for their feedback and time. We are happy that the work is considered well-written, with strong experiments, clear and informative figures, and a detailed appendix. We are pleased that the evaluation on a private dataset is appreciated and provides strong motivation for the work and that the inclusion of expert information is interesting.
>
> In our rebuttals, alongside our point-by-point responses, we present **three new experiments**.
>
> ### **Suggested improvements**
>
> **Q1: Discussion of related works in LLMs from a Bayesian perspective**
>
> As you note, numerous works discuss in-context learning from a Bayesian perspective. Our work is unique in that we empirically estimate the internal prior and posterior distributions of an in-context learner. With this, we use MCMC methods to compare the in-context posterior with a separately calculated posterior. This allows us to question whether an LLM is performing Bayesian inference on *linear modelling of real-world numerical tasks*. This is more relevant to our work than the previous natural language or theoretical perspectives. This contribution also provides a complete decision criterion to determine whether in-context learning or AutoElicit is more performant for a task.
>
> We agree that this is an important area of the literature. We were restricted by the page length requirements of the submission and will include a complete description in the main section of the final paper with your suggestions.
>
> **Q2: Discussion on the interpretability of the priors**
>
> We use 100 mixture components in our prior to mitigate the effects of LLM hallucination, which is important in healthcare tasks. The prior distribution can be understood by visualising histograms of the parameter samples from the final mixture (linked below in a modified Figure 15):
>
> **FIGURE:** https://imgur.com/a/gRRVmUz
>
> As the density of these distributions are used for the prior directly, this provides a complete picture. For linear models, each feature's prior corresponds to its direction and importance when predicting the target. On the synthetic task (row 1), the prior distributions concentrate on the parameters that we expect. For the UTI task (row 2), the LLM provided prior distributions that suggest Feature 2 was negatively predictive of a positive UTI, whilst Feature 10 is a strong positive predictor. For Breast Cancer (row 6), the LLM produced two modes in its Feature 2 prior. Here, the LLM was either unsure of its belief, and so the elicited prior changed between task descriptions, or the LLM was unsure of the feature’s direction of correlation *but guessed that the feature was important*.
>
> A chain-of-thought LLM like Deepseek-R1 could help us to better understand the elicitation process. We also found that using LLMs for predictions directly (in-context learning) underperforms in comparison to our approach.
>
> **Q3: Discussion on the number of mixture components**
>
> We investigate, in **Rebbutal 2 for Reviewer 2 (3Pnd), in section Q3**, how the posterior performance depends on the number of prior components. Here, we significantly reduce the number of mixture components whilst retaining the posterior accuracy and mean squared error.
>
> **Q4: *New experiment* evaluating an uninformative mixture of random Gaussians**
>
> Based on your suggestion, we compare our results to a new uninformative baseline. We sample 100 values from a standard normal distribution ($\mu \sim N(\mu | 0,1)$), and we use these as means of 100 Gaussian distributions ($\theta \sim N(\theta | \mu, 1)$). With these distributions we construct a mixture prior. As we keep the standard deviation at 1 for each component, the resulting prior has a greater range in the parameter space. The figure linked below shows the results:
>
> **FIGURE:** https://imgur.com/a/STNjyEG
>
> This new uninformative prior leads to no noticeable difference in the results for the synthetic task, Diabetes task, and Breast Cancer task. However, on the UTI task, we see marginally better accuracy than the previous uninformative prior, and on the Heart Disease and Hypothyroid tasks, we see greater improvements. Despite this, DeepSeek-R1-32B-Q4 or GPT-3.5-turbo continue to provide more informative priors. Therefore, the posterior improvement is not due to the complexity of the prior.
>
> **Q5: Clarity on the predictive performance**
>
> Predictive performance refers to either the posterior accuracy or mean squared error. To calculate this, we sample 5 chains of 5000 posterior parameter samples using MCMC methods. We use these sampled posterior parameters to predict the labels on our test set and calculate the metrics. This provides a distribution of performance.
>
> **Thank you for taking the time to review our work. We will implement the suggested improvements to the final manuscript. If we have answered your questions, then we would appreciate you considering raising your score. If anything is still unclear, we are happy to discuss further.**

---

### Official Review · Reviewer_3Pnd · 2025-03-15

**Overall Recommendation:** 4

**Summary:**

Linear predictive models remain valuable for applied researchers. However, principled Bayesian approaches to such models require the specification of some prior. What prior should researchers use for different parameters? Good priors may be informed by expert information, but such expert knowledge can be hard to come by. The authors explore the possibility of using language models to elicit such priors. They develop a new method, AutoElicit, to automatically extract priors from a range of LLMs and compare against direct prediction and ICL elicitation from models. Generally, they find that the priors elicited from their method yield better downstream predictive performance on a range of medical tasks (and one synthetic predictive task).

**Claims And Evidence:**

The paper is very well-written and thorough. I particularly appreciate the authors' extensive Appendix. They clearly ran a wide range of experiments that probe the characteristics of AutoElicit against ICL. I particularly appreciate the authors' exploration of other LLMs --- their work raises interesting questions about which model to use, when, based on costs of elicitation --- as well as their analysis of the potential impact of memorization on results.

What I was less clear about, though, is:
- (1) Where AutoElicit stands in relation to other prior elicitation methods in terms of performance
- (2) The value / ease of incorporating expert information
- (3) The role of the diversity of task descriptions

On (1), while I admire the authors' extensive literature review, I was a bit disappointed to not see direct head-to-head comparisons with some alternate elicitation methods. For instance, the authors mention Zhu and Griffiths --- how does such a method compare on something like the UTI dataset? It would be good to be clearer on why alternate methods were not considered in the empirical section?

For (2), the authors claim that their method is a way to also incorporate expert knowledge into prior elicitation. However, the UTI example they gave -- to my sense? -- had more "commonsense medical knowledge" that was added (e.g., UTIs lead to more urination at night). I think it makes sense that GPT-3.5 wasn't really impacted by this; I wouldn't say that's real "expert" knowledge per say? Can the authors sway parameters even more with genuine, nice expert knowledge? This need not be in this current submission in my opinion, but is something I walked away curious about and did not feel the current paper really demonstrated.

And on (3), I was quite confused on the task description resampling. Based on the Appendix, the number seems to make quite a big difference! But what is the character of these different descriptions? Can you add some examples to the Appendix? How different are they really? Or is the "boost" from just resampling?

**Essential References Not Discussed:**

I would encourage the authors to look at:

Large Language Models Are Latent Variable Models: Explaining and Finding Good Demonstrations for In-Context Learning https://proceedings.neurips.cc/paper_files/paper/2023/hash/3255a7554605a88800f4e120b3a929e1-Abstract-Conference.html -- looks at LLMs and ICL from a Bayesian lense

Automated Statistical Model Discovery with Language Models https://arxiv.org/abs/2402.17879 which also looked at the use of LLMs in more classical statistical modeling

MPP: Language Models as Probabilistic Priors for Perception and Action https://arxiv.org/pdf/2302.02801 --- an early work using LLMs as priors (though not for linear modeling problems as these authors consider).

**Experimental Designs Or Analyses:**

As noted above, I think the authors did a very thorough job in a range of experiments -- however, I am particularly curious about the role of the task descriptions versus potentially

**Methods And Evaluation Criteria:**

Generally I think the evaluation was sound, however,  as noted above, I do not completely understand why the number of task descriptions has such a big role (Fig 14). Is this really from the task descriptions, or something else? I see the authors set the temperature to 0.1? Why generally so low, if part of the role is to get diversity in parameters?

I also came away with some confusion over whether the priors were semantically sensible. For instance, in Fig 15, some parameters' priors shifted around quite a bit -- others not. Can the authors add more qualitative analyses into what changed and whether it is semantically-relevant to the task at hand?

**Other Comments Or Suggestions:**

Fig 17 says "OpenAI models" but includes DeepSeek?

**Other Strengths And Weaknesses:**

I greatly admire the authors' effort to make the code easily runnable (A.1) and their detailed description of how to run the method (A.7.6). I think this will be valuable to the broader community. Well done!

For future (not here), I would be keen to see how actual experiments -- e.g., doctors from the UTI study? -- interpret the parameters and assess the value of AutoElicit.

**Questions For Authors:**

I raised my questions above --- particularly around the task descriptions and some deeper parameter analyses.

**Relation To Broader Scientific Literature:**

I think the authors did a generally good job of situating their work in relation to other literature. I think the experiments around "do LLMs do Bayesian inference" though did not provide much substantial value ontop of what's already been looked at in the literature. I'd encourage the authors to focus principally on the role of prior elicitation, which seems to be the more novel direction here.

I would have liked to see though more empirical comparison to at least one alternate method for prior elicitation from LLMs. How does AutoElicit compare?

**Theoretical Claims:**

I believe the theory is appropriate but am not 100% confident in my assessment.

---

> ### Author Rebuttal · Authors · 2025-03-31
>
> We appreciate the reviewer's kind words and time. We are pleased that our work was considered well-written with thorough experiments and appendices. We are happy that the extensive literature review, discussion of elicitation costs, and LLM memorisation experiments were appreciated.
>
> In our rebuttals, alongside our point-by-point responses, we present **three new experiments**.
>
> ### **Suggested improvements**
>
> **Q1: Further comparisons to a baseline**
>
> Zhu & Griffiths (2024) is not directly comparable to our work, as they do not provide a framework for eliciting the parameters of a predictive model. The most comparable work, Gouk & Gao (2024) (cited), generates synthetic data using an LLM to include in the training of linear *classification* models. To compare with our method, we extend their ideas to *regression* tasks and present new experimental results.
>
> These are presented in our response to **Reviewer 1 (xAxP) in section Q2**, along with a detailed description of the baseline. In particular, Gouk & Gao’s method is susceptible to LLM data memorisation, putting it at risk of artificially providing improved posterior results on public datasets. To better compare the methods, we have split the results by the private and public tasks. We find that our approach yields improved posterior accuracy and mean squared error over Gouk & Gao’s method.
>
> **Q2: Using more challenging expert information**
>
> This is an interesting point. A synthetic example of an LLM using completely new expert information is given in the last paragraph of Appendix A.11 and Figure 13 (linked below).
>
> **FIGURE:** https://imgur.com/a/QSZl2ZO
>
> This demonstrates that giving increasingly more detailed information about the task (provided as natural language, Appendix A.11) allowed the LLM to update the elicited distributions correctly. We will lengthen the discussion in Section 5.3 and Appendix A.11 of the final paper to include this point.
>
> **Q3: *New experiment* on the variation of the task descriptions and number of mixture components**
>
> We will include examples of rephrased task descriptions in the appendix. We used a temperature of 0.1 to elicit the prior distributions so that the LLM reliably generated parameters of a Gaussian distribution and to mitigate the effects of hallucination, as we focus on healthcare tasks. When rephrasing the task descriptions, we used a higher temperature of 1.0 to get diverse descriptions. This is why we see variations in the priors elicited. An example rephrased system role is:
>
> > You are a simulator of a logistic regression predictive model for … Here the inputs are values from sensors around a home and the output is the probability of the presence of a urinary tract infection. Specifically, the targets are …
>
> To:
>
> > You function as a logistic regression prediction model for … Inputs include sensor readings from a home, and the output is the probability of a urinary tract infection. The targets are …
>
> The meaning of the description remains the same, but the rephrasing mitigates against changes in the priors elicited based only on the language used in the prompt.
>
> To supplement this, we calculate the posterior performance with varied numbers of mixture components in our prior (linked below):
>
> **FIGURE:** https://imgur.com/a/E3hE9Na
>
> We see little variation (except for Breast Cancer with Deepseek) in the accuracy or mean squared error as we increase the number of components in the prior. This suggests our framework is robust to the number of task descriptions. We hypothesise that a greater number of mixture components makes the framework more resilient to LLMs that frequently hallucinate, especially on tasks with large numbers of features (such as Breast Cancer).
>
> In Appendix A.12, we showed that the density of the priors changed with the number of mixture components, however, this has small effects on the posterior performance. This is likely because the sample space that provides good posterior performance is covered by the majority of elicited components, whilst the large changes in Appendix A.12 are due to more unique components.
>
> **Q4: Discussion on the interpretability of priors**
>
> In our response to **Reviewer 3 (fJAR) in section Q2**, we provide examples of interpreting the priors elicited from GPT-3.5-turbo. As suggested, it would be interesting to explore how these priors are understood by clinicians and if they are used separately from the predictive modelling. The points you have raised are valuable. This is future work we are currently organising with clinicians.
>
> **Related works**
>
> Thank you for the helpful suggestions. We will include these references with discussions in the final version.
>
> **Thank you for taking the time to review our work. We will implement the suggested improvements to the final manuscript. If we have answered your questions, then we would appreciate you considering raising your score. If anything is still unclear, we are happy to discuss further.**

---

### Official Review · Reviewer_xAxP · 2025-03-17

**Overall Recommendation:** 3

**Summary:**

The paper considers a prior elicitation method based on query large language models, rather than human experts, for the purposes of fitting Bayesian linear models. They make a distinction between the explicitly elicited priors supplied by their method, and the implicit priors used by the LLM when doing in-context learning. They have a number of insights related to how LLM do not exhibit consistent Bayesian reasoning, and the how expliticly elicited priors can lead to improved sample efficiency outperform in-context learning.

**Claims And Evidence:**

The paper makes three main claims:
* The introduction of a new algorithm for eliciting priors over linear models from LLMs, which leads to better loss values---particularly with small training sets. This is substantiated by experimental results. However, I would note that the algorithm they present is essentially the same as the one proposed in Selby et al. (2024), but applied to a different set of modelling problems.
* When comparing with in-context learning, it is found that LLMs inconsistently approximate Bayesian inference, and that their implicit priors are less informative than those that are explicitly elicited.
* Bayes factors are claimed to be a useful tool for comparing ICL and linear models with LLM-based priors. This is validated experimentally.

**Essential References Not Discussed:**

N/A

**Experimental Designs Or Analyses:**

The main issue I can see with the experimental evaluation is the lack of comparison with the very related method of Gouk & Gao (2024), who also consider eliciting priors for linear models from LLMs. Other than that, the experiments are quite well though out. The use of methods to determine leakage of datasets in the LLM pre-training, and also a private dataset, is a particularly positive aspect of the setup.

**Methods And Evaluation Criteria:**

The method and evaluation criteria mostly make sense: using LLMs as a generic knowledge base has ample motivation, and model selection based on Bayes factors is well-established.

**Other Comments Or Suggestions:**

N/A

**Other Strengths And Weaknesses:**

N/A

**Questions For Authors:**

N/A

**Relation To Broader Scientific Literature:**

Compared to the two most related previous pieces of work in this area (Gouk & Gao (2024) and Selby et al. (2024)), this work provides more analysis of how the explicitly elicited priors compare to the implicit priors used during in-context learning.

One aspect that I do not think is sufficiently highlighted in the paper is that the proposed method is essentially the same as Selby et al. (2024). The difference between this previous work and the current paper lies only in the types of tasks used during evaluation---Selby et al. consider imputation and data generation problems, rather than linear modelling problems.

**Theoretical Claims:**

N/A

---

> ### Author Rebuttal · Authors · 2025-03-31
>
> We would like to thank the reviewer for their comments and constructive feedback to improve our work. We are happy the work was found to be well-written, easy to follow, and well motivated. We are pleased that the use of strong experiments, evaluation of LLM memorisation, and use of our privately collected clinical dataset were appreciated.
>
> In our rebuttals, alongside our point-by-point responses, we present **three new experiments**.
>
> ### **Suggested improvements**
>
> **Q1: Methodological comparison to Selby et al. (2024)**
>
> Thank you for your question. We list key differences between our work and Selby et al. (2024) (cited), highlighting where the novelty of our approach lies. We will improve our description of this work in the final version of the manuscript:
> - Selby et al.’s core focus is on data imputation; however, they show preliminary results for eliciting distributions and compare these to human experts.
> - They use these in a single predictive task. However, their method elicits a prior *over the targets* directly, much like with in-context learning, and not *over the parameters* of a model as in our work. This means that Selby et al. do not update a model using available data and are therefore not training any predictive models using the priors. This also makes Selby et al.’s work incompatible as a baseline.
> - Selby et al. do not perform repeated sampling of the LLM, which we hypothesise mitigates hallucination.
> - Selby et al. do not combine experts and LLMs, an important aspect of human-AI interaction. We achieve this by allowing experts to provide natural language information in task descriptions, simplifying current prior elicitation methods. In this case, our elicited distributions contain knowledge from both the experts and the LLM’s training.
> - We also present a grounded model selection criterion to decide between AutoElicit and in-context learning.
>
>
> **Q2: *New experiment* with Gouk & Gao (2024) as a baseline**
>
> A more compatible baseline presented by Gouk & Gao (2024) (cited) generates synthetic data using an LLM to provide an additional likelihood term when training linear *classification* models. This method uses an LLM to generate feature values before labelling them with zero-shot in-context learning. These generated samples are provided alongside real data during the training of a linear predictive model. Therefore, Gouk & Gao’s method is susceptible to LLM data memorisation, putting it at risk of artificially providing improved posterior results on public datasets.
>
> We compare our results to this baseline and extend their ideas to *regression* tasks to compare with all the datasets we test. This previous work was described in our manuscript but we now provide new experimental results. The figure (linked below) will be included in the final manuscript:
>
> **FIGURE:** https://imgur.com/a/3sM2baQ
>
> This figure is split by publicly available tasks (four right-most plots) and privately available tasks (two left-most plots). In these experiments, our approach provides significantly greater posterior accuracy and lower posterior mean squared error for both private datasets.
>
> In the public tasks, we see that the performance of Gouk & Gao’s approach varies considerably by LLM and task. It underperforms compared to the *uninformative* prior on five occasions. On the two occasions it outperforms AutoElicit (ours), it achieves surprisingly good accuracy or mean squared error (Heart Disease with GPT-4-turbo and Diabetes with GPT-3.5-turbo), suggesting that it might be reproducing parts of the public datasets rather than generating new samples. Given our analysis in Appendix A.8 showing the memorisation ability of LLMs, and Gouk and Gao’s performance on private data, we suspect the performance of this approach is artificially improved by reproducing the true dataset. Since for AutoElicit, we prompt the language model for a prior distribution over the parameters of a predictive model, using only the feature names and a description of the dataset, the memorisation of data points is unlikely to have had an impact on our method's results (discussed further in Appendix A.8.1).
>
> In both of the private datasets (two left-most plots), where there is no risk of LLM memorisation, AutoElicit results in improved posterior accuracy and mean squared error. On the synthetic task, our improvement is orders of magnitudes better, and on the UTI task, our method significantly increases accuracy at all training sizes. In both cases, the approach proposed by Gouk & Gao underperforms compared to an *uninformative* prior.
>
> We will include these new results in the final version of the paper, as it provides further context to our approach.
>
> **Thank you for taking the time to review our work. We will implement the suggested improvements to the final manuscript. If we have answered your questions, then we would appreciate you considering raising your score. If anything is still unclear, we are happy to discuss further.**

---

### Decision · Program_Chairs · 2025-05-01

**Decision:**

Accept (poster)

**Comment:**

This paper outlines a method for using LLMs to elicit Bayesian priors for linear models. While building on previous work, the reviewers consider AutoElicit to make a nice contribution to the area, with a compelling presentation, and strong empirical results. The author rebuttal cleared up a number of questions raised by the reviewers, and the inclusion of new empirical results (placed nicely in context) makes the paper stronger. That said, one reviewer remains somewhat concerned about the question of interpretability, and the authors response was, in my opinion, somewhat weak on this score. Still, the overall contribution is solid.

The reviewers had a number of valuable suggestions on the paper, which I strongly encourage the authors to address in any revision of the paper to solidify its contribution—this includes ensuring the material contained in the rebuttal is incorporated (e.g., new experiments).